# Seek-CAD: A Self-refined Generative Modeling for 3D Parametric CAD Using Local Inference via DeepSeek

**Xueyang Li**[1,2]*, **Jiahao Li**[1]*, **Yu Song**[1], **Yunzhong Lou**[1], **Xiangdong Zhou**[1]✉

[1]College of Computer Science and Artificial Intelligence, Fudan University, Shanghai, China
[2]School of Information and Intelligent Science, Donghua University, Shanghai, China
`{xueyangli21, 25113050261, songy23}@m.fudan.edu.cn`
`{yzlou20, xdzhou}@fudan.edu.cn`

## Abstract

The advent of Computer-Aided Design (CAD) generative modeling will significantly transform the design of industrial products. The recent research endeavor has extended into the realm of Large Language Models (LLMs). In contrast to fine-tuning methods, training-free approaches typically utilize the advanced LLMs, thereby offering enhanced flexibility and efficiency in the development of AI agents for generating CAD parametric models. However, the lack of a mechanism to harness Chain-of-Thought (CoT) limits the potential of LLMs in CAD applications. The Seek-CAD is the pioneer exploration of locally deployed inference LLM DeepSeek-R1 for CAD parametric model generation with a training-free methodology. This study is the investigation to incorporate both visual and CoT feedback within the self-refinement mechanism for generating CAD models. Specifically, the initial generated parametric CAD model is rendered into a sequence of step-wise perspective images, which are subsequently processed by a Vision Language Model (VLM) alongside the corresponding CoTs derived from DeepSeek-R1 to assess the CAD model generation. Then, the feedback is utilized by DeepSeek-R1 to refine the initial generated model for the next round of generation. Moreover, we present an innovative 3D CAD model dataset structured around the SSR (Sketch, Sketch-based feature, and Refinements) triple design paradigm. This dataset encompasses a wide range of CAD commands, thereby aligning effectively with industrial application requirements and proving suitable for the generation of LLMs. Extensive experiments validate the effectiveness of Seek-CAD under various metrics.

## 1 Introduction

The CAD parametric model (also called design history), as a crucial role in Computer-Aided Design (CAD), indicates the design logic of 3D CAD models, where its command and parameter sequences can be quickly edited to create or modify the shape of a 3D object Wu et al. (2021); Jones et al. (2023). However, constructing a parametric CAD model from scratch is time-consuming and hinders the development of automation in industrial manufacturing. Hence, it attracts much attention on generative CAD modeling recently, including many interesting applications such as CAD parts assembly Jones et al. (2021); Willis et al. (2022); Wu et al. (2023), shape parsing Li et al. (2022); Ren et al. (2022), CAD parametric model generation Wu et al. (2021); Xu et al. (2023); Li et al. (2025c), and cross-modal CAD generation (e.g., Point cloud-to-CAD, Text-to-CAD, and etc.) Khan et al. (2024a); Li et al. (2024a); Dupont et al. (2024); Li et al. (2024c;b); Ma et al. (2024); Li et al. (2025d); Zhang et al. (2025b); Khan et al. (2024b). Under the current trend of LLMs and VLMs performing outstandingly in computer vision tasks Wang et al. (2023); Schumann et al. (2024); Wang & Ke (2024); Huang et al. (2024); Zhu et al. (2024); Bensabath et al. (2024), integrating LLMs and VLMs in generative CAD modeling will pave the way for future innovations in smart design systems.

---

*Equal contribution

Fine-tuning is a commonly used method for adapting general LLMs to domain-specific applications. In contrast to fine-tuning approaches, training-free approaches typically utilize advanced LLMs i.e. GPT-4o Achiam et al. (2023), thereby offering enhanced flexibility and efficiency in the creation of AI agents for generating CAD parametric models Yuan et al. (2024); Alrashedy et al. (2025). However, the primary drawback is the absence of a mechanism to harness Chain-of-Thought (CoT), which limits the potential of LLMs in CAD applications. Inspired by DeepSeek-R1 Guo et al. (2025) with advanced capability of reasoning, we deploy a DeepSeek-R1-32B-Q4 locally without training or finetuning as the backbone of our approach to explore its capability for generative CAD modeling.

Specifically, we present Seek-CAD, a training-free generative framework for CAD modeling based on DeepSeek-R1-32B-Q4. By employing a retrieval-augmented generation (RAG) strategy on a local CAD code corpus, Seek-CAD produces Python-like CAD code for CAD parametric model generation. To ensure that the generated code aligns with prompt descriptions and faithfully encodes geometric features (e.g., *fillet*, *chamfer*) and constraints (e.g., *tangency*, *orthogonality*), we introduce a step-wise visual feedback mechanism that guides and refines the modeling process. In particular, we render step-wise perspective images that visually capture each stage of the CAD modeling process. These images are then evaluated using Gemini-2.0 Team et al. (2024) to assess their alignment with the chain-of-thought (CoT) from DeepSeek-R1-32B-Q4. The resulting feedback is incorporated to iteratively refine the initial code and parameters.

Furthermore, we propose a novel CAD design paradigm called SSR (Sketch, Sketch-based feature, and Refinements), where each model is represented as a sequence of SSR triples, each consisting of a sketch, a sketch-based feature (e.g., *extrude* and *revolve*), and optionally, refinement features (e.g., *chamfer*, *fillet* and *shell*). Complex shapes are constructed through boolean operations across SSR units. To support refinement features, we introduce a simple yet effective reference mechanism, termed *CapType* (Figure 3), which establishes explicit links between topological primitives in the sketch and their resulting primitives generated during modeling. To evaluate our approach, we construct a new CAD dataset of 40k samples following the SSR modeling paradigm. The dataset covers diverse CAD features not included in existing datasets, and each sample is paired with a textual description generated by GPT-4o Achiam et al. (2023). For further details, please see Section A.2. The datatset has been released publicly and can be acessed in `https://github.com/Sunny-Hack/Seek-CAD`.

In summary, our key contributions are as follows: (i) We present Seek-CAD, a training-free framework leveraging the locally deployed DeepSeek-R1. It incorporates a self-refinement capability through a sequential visual and CoT feedback mechanism, which enhances the generative modeling of CAD designs and significantly contributes to the effective generation of diverse CAD parametric models. (ii) We present an innovative SSR design paradigm, which serves as an alternative to the conventional SE paradigm and demonstrates enhanced suitability for the generation of complex CAD models. (iii) Experimental results demonstrate that Seek-CAD can generate diverse and complex parametric CAD models with high geometric fidelity, enabling precise parametric control while supporting the generation of diverse CAD models.

## 2 RELATED WORK

### 2.1 GENERATIVE CAD MODELING

Generative CAD modeling has made a significant step recently Xu et al. (2022; 2024b); Jayaraman et al. (2021); Dupont et al. (2024); Guo et al. (2022), notably with transformer-based models, which treat CAD commands as parametric sequential data for learning and generation with a feed-forward or an auto-regressive strategy Wu et al. (2021); Xu et al. (2022; 2023); Ganin et al. (2021). Besides, diffusion-based Ho et al. (2020) methods have also been adopted to achieve the controllable generation or reconstruction of parametric CAD sequences Ma et al. (2024); Zhang et al. (2025a). Meanwhile, reinforcement learning offers another effective paradigm for generating CAD models Li et al. (2025a). Mamba-CAD Li et al. (2025c) makes a step forward to handle longer parametric CAD sequences to generate complex CAD models. As a CAD model contains explicit parametric commands, which implicitly indicate its design logic and shape geometry, it can be seen as a kind of multi-modal data inherently, which also raises many applications of generating or reconstructing CAD models from point clouds, images, and texts Li et al. (2024b); You et al. (2024); Dupont et al. (2024); Khan et al. (2024a); Li et al. (2024c). These methods mainly focus on the SE (Sketch-Extrusion) paradigm,

which supports only a limited set of simple CAD operations, making it incapable of generating diverse and complex CAD models that meet real-world design requirements. Instead, Seek-CAD adopts the SSR paradigm, which enables the inclusion of diverse CAD commands (similar to the complex feature set in WHUCAD Fan et al. (2025)) such as *fillet*, *chamfer*, and *shell*, supporting the creation of more complex CAD models, which is closer to industrial requirements, which is closer to industrial requirements. From another perspective, Seek-CAD is a self-refined framework for generative CAD modeling without any training or finetuning, which is also different from these efforts under a training strategy.

## 2.2 LLMs FOR CAD

With the success of LLMs in Computer Vision (CV) tasks Tan et al. (2024); Naeem et al. (2023); Shao et al. (2024); Ghosh et al. (2024), many interesting applications of leveraging LLMs for CAD-related tasks have raised much attention recently.Text2CAD Khan et al. (2024b) generates parametric CAD models from natural language instructions using a transformer-based network Vaswani et al. (2017) with an annotation pipeline for CAD prompt using Mistral Chaplot (2023) and LLaVA-NeXT Liu et al. (2024). CAD-MLLM Xu et al. (2024a), a unified system to utilize a LLM to align multimodal inputs with parametric CAD sequences for generating CAD models. CAD-assistant Mallis et al. (2024) uses a Vision-Large Language Model (VLLM) with tool-augmented planning to iteratively generate and adapt CAD designs via Python API of FreeCAD. FI2CAD Ocker et al. (2025) introduces a multi-agent system, a LLM-based MAS architecture for CAD development processes that mimics an engineering team to automatically generate and refine parametric CAD models with human feedback. CAD-Llama Li et al. (2025b) enables pretrained LLMs to generate parametric 3D CAD models through the adaptive pretraining on Structured Parametric CAD Code (SPCC). Unlike these methods, our approach mainly focuses on a training-free strategy with the self-refined capability for generative CAD modeling.

The most related to our Seek-CAD are other two training-free frameworks, 3D-PreMise Yuan et al. (2024) and CADCodeVerify Alrashedy et al. (2025). 3D-Premise enhances CAD code refinement by supplying GPT-4 Achiam et al. (2023) with an image of a whole object and its initial description, allowing it to identify and correct discrepancies between the intended design and the generated CAD code. Unlike 3D-Premise, CADCodeVerify prompts GPT-4 Achiam et al. (2023) to generate and answer a set of questions based on the initially provided description of a 3D CAD object, to adjust any discrepancies between the generated CAD model and the description. Compared to them, Seek-CAD differs in the following aspects: (i) Seek-CAD uses DeepSeek-R1-32B-Q4 (open-source reasoning model) for local deployment, integrated with a CAD code corpus via a retrieval-augmented generation framework, instead of relying on GPT-4. (ii) Previous research evaluates object renderings in VLMs by focusing only on final CAD forms, overlooking intermediate phases, which limits feedback for complex models. Seek-CAD addresses this with a step-wise feedback mechanism, showing both final and intermediate shapes to enhance VLM feedback. (iii) Previous methods use predefined question templates to evaluate VLM alignment with descriptions and images. Seek-CAD, however, guides VLMs to assess alignment between DeepSeek-R1's chain-of-thought (CoT) and step-wise images, as CoT effectively illustrates the design logic, enabling clearer VLM understanding (Sec 3.2).

## 3 SEEK-CAD FRAMEWORK

In this section, we introduce the Seek-CAD pipeline, which integrates a local inference process (Sec 3.1) and a step-wise visual feedback strategy (Sec 3.2) to generate and progressively refine CAD models, based on visual alignment signals derived from step-wise renderings. The framework of Seek-CAD is illustrated in Figure 1.

### 3.1 LOCAL INFERENCE PIPELINE

LLMs are widely applied across various domains through training or finetuning, both of which require substantial computational resources. In contrast, performing local inference with LLMs can significantly reduce the requirement for computational resources. Inspired by the advancements in reasoning capabilities of DeepSeek-R1, we make a step forward to explore its potential for CAD generative modeling without additional training or finetuning.

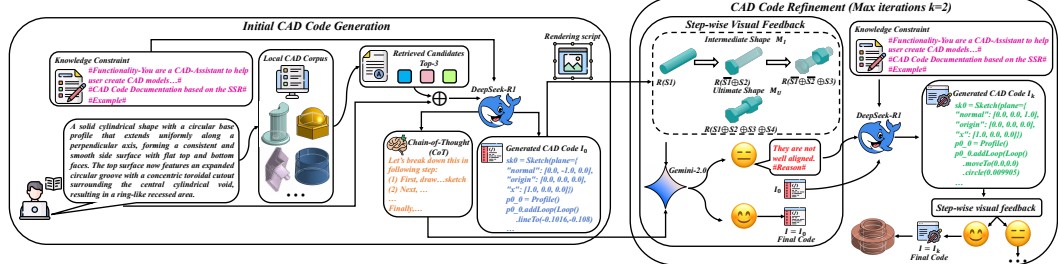

Figure 1: The overview of our Seek-CAD framework. The whole pipeline can be divided into two parts consisting of "Initial CAD Code Generation" and "CAD Code Refinement", which are both embedded with a knowledge constraint depicted in Sec. 3.1 to guide DeepSeek-R1 to generate CAD code following the SSR paradigm (Sec. 4). For the first part, a given query $T$ is enhanced by conducting RAG on a local CAD corpus that consisting 10, 000 CAD models. Next, Top-3 retrieved candidates would be concatenated with $T$ to trigger DeepSeek-R1 to generate an initial CAD code $I_0$. For the second part, $I_0$ would go through the Step-wise Visual Feedback with CoT to have the iteration refinement. To achieve this, we first utilize a rendering script $R(*)$ to obtain step-wise images of $I_0$, which can represents the intermediate and ultimate shape of the object ($M_I$, $M_U$) simultaneously. $\oplus$ denotes the concatenation of SSR triplets, where each triplet, represented by $S_i$ (Sec. 4), is rendered along with all its preceding triplets to preserve the correlations between object entities. (More details in Sec. 3.2). Next, the step-wise images are fed into Gemini-2.0 to assess their alignment with the CoT from DeepSeek-R1. This feedback determines whether the current code $I_k$ is reasonable. In practice, we set $k = 1$ as the maximize iterations of code refinement.

**(1) Pipeline Definition.** This pipeline takes the text $T$ as the input. It is a statement of how to design a CAD model step by step, or it can also be a description of the geometric appearance of objects. The goal can be defined as: $I_0 \sim P(I_0 \mid T; H)$ that maps the input text $T$ to an initial set of CAD code $I_0$ represented in the SSR paradigm (Sec 4), where $H$ denotes our Seek-CAD.

**(2) Knowledge Constraint.** Similar to other LLMs, DeepSeek-R1 may exhibit hallucinations Sriramanan et al. (2024); Huang et al. (2025); Jiang et al. (2024), occasionally generating CAD parametric models that deviate from the SSR paradigm. To address it, we propose the knowledge constraint $Cons = (\Phi, \mathcal{D}, \mathcal{E})$ as the system prompt to make DeepSeek-R1 generate CAD code following the SSR paradigm, with the constraint $Cons$ consisting of three parts: $\Phi$ specifies its functionality, $\mathcal{D}$ documents the SSR schema, and $\mathcal{E}$ provides an example pairing a textual description with SSR-based CAD code, as shown in Figure 6 of Appendix A.1.

**(3) Retrieval Augment Generation (RAG).** In order to augment the recognition of Seek-CAD within the SSR paradigm, we construct a SSR-based CAD corpus $\mathcal{C}_{SSR} = \{(d_i, c_i)\}_{i=1}^{N}$ to equip Seek-CAD with the capability to perform RAG. $d_i$ is the textual description and $c_i$ its corresponding CAD code, with $N = 10,000$ CAD models. As usual settings of conducting RAG, we select a hybrid search combining a vector-based and full-text strategy. Specifically, for a given $T$, the similarity score is computed by:

$$g_i^{\text{final}} = \lambda \cdot g_i^{\text{vec}} + (1 - \lambda) \cdot g_i^{\text{full}}, \quad \lambda \in [0, 1], \tag{1}$$

where $g_i^{\text{vec}} = \cos(z_T, z_i)$, $z_T = \mathbf{e}_{\text{vec}}(T)$, $z_i = \mathbf{e}_{\text{vec}}(d_i)$. $\mathbf{e}_{\text{vec}}$ denotes an embedding model, bge-m3 Chen et al. (2024). $g_i^{\text{full}} = \mathbf{U}_{index}(T, d_i)$ is calculated with a traditional inverted index without vectorization. The top-$k$ candidates are selected as $\mathcal{R}_T = \text{TopK}(g_i^{\text{final}})$. Practically, we set $\lambda = 0.3$, top-$k = 3$.

**(4) Initial CAD Code Generation.** To incorporate the retrieved content into the input context, we make each retrieved pair $(d_j, c_j) \in \mathcal{R}_T$ concatenated with $T$. By integrating the knowledge constraints $Cons$, the initial CAD code $I_0$ can be obtained as follows:

$$I_0 \sim P(I_0 \mid T \oplus (d_j, c_j), Cons). \tag{2}$$

The initially generated CAD code $I_0$ is occasionally subject to compilation failures with the geometry kernel (e.g., PythonOCC Paviot (2022) utilized in this study), which are attributed to syntax errors $E$. For remediation, a pattern template $Q$ is employed to automatically rectify $E$ in $I_0$, addressing issues such as mismatched parentheses and incorrect capitalization of variable names, depicted as: $I_0 \sim P(I_0 \mid Q)$.

**(5) CAD Code Refinement.** Upon addressing the aforementioned syntax errors, we employ the geometry kernel to directly render $I_0$, thereby acquiring its sequential perspective images (Sec 3.2). Subsequently, the sequential perspective images, in conjunction with the CoT obtained from DeepSeek-R1, are input into Gemini-2.0 for the purpose of evaluating their alignment. Finally, the step-wise visual feedback (Sec 3.2), $F_{call}$, would be delivered back to DeepSeek-R1 again to refine $I_0$ to get final CAD code $I$. This process may iterate $N$ times based on whether $F_{call}$ is positive or negative, To make it clear, we denote $F$ as an indicator to represent the statement of $F_{call}$. For example, if $L = 1$, it means $F_{call}$ is positive, that is, the CoT and step-wise perspective images are well matched and do not require any further modification, which can be defined with:

$$I = \begin{cases} \{I_m\}_{m=0}^k & L = 1 \\ I_k \sim \{P(I_k \mid I_{k-1}, F_{call}, Cons)\}_{k=1}^N & L = 0 \end{cases}. \tag{3}$$

Analogous to the initial CAD code $I_0$, which bears the potential for syntax errors, a syntax check shall also be executed on $I_k$ during each iterative step, as delineated in: $I_k \sim P(I_k \mid Q)$. In practice, the maximum iteration step $N = 2$ is established in our work to avert superfluous adjustments, which may arise from the hallucinations of Gemini-2.0. Further elaboration on the iteration step within the refinement stage can be found in the "Refinement" section of Sec 5.2.

## 3.2 STEP-WISE VISUAL FEEDBACK WITH CoT FOR REFINEMENT

A significant capability of reasoning LLMs lies in their ability to refine and optimize initial outputs in response to feedback. The self-refined strategy has been adopted in CAD parametric model generation Yuan et al. (2024); Alrashedy et al. (2025). As CAD command sequences can be easily rendered into the perspective image of an object by using geometry kernel tools like PythonOCC, this can shift the aligned judgment from "Command-Description" to "Image-Description", which can be well handled by VLMs. In this study, we propose a novel step-wise visual feedback (SVF) strategy, which leverages not only the imagery of the object's final form but also retains visuals depicting the intermediate configurations throughout the entire construction process. Additionally, it incorporates the CoTs from DeepSeek-R1 to guide the VLM during its evaluation.

**(1) Obtain Step-wise Images.** Specifically, the initial generated CAD code $I_0$ is converted into a sequence ($S_{seq}$) of $n$ SSR triplets, $S_i$ (Recall Sec 4), as defined with $I_0 \rightarrow S_{seq} = [S_1, S_2, \cdots, S_n]$. Next, we leverage the rendering script $R(*)$ (based on PythonOCC) to subsequently compile this sequence $S$ to generate their corresponding perspective images. To better illustrate the construction process from a visual perspective, we capture both the intermediate object shapes, $M_I$, and the ultimate object shape, $M_U$. To capture $M_I$, for each step $k \in \{1, \ldots, n\}$ in the construction sequence $S_1 \oplus S_2 \oplus \cdots \oplus S_k$, the corresponding image in $M_I$ is rendered by $R(*)$. To maintain the correlations between entities of the object and make them visualized clearly, this rendering process would highlight the entity from the current SSR triplet $S_k$ while entities from all prior SSR triplets $S_j$ (where $j < k$) are hidden as $\bar{S}_i$. To achieve this, we adjust the transparency in $R_{(*)}$ to ensure that the current shape of $S_k$ to the $S_j$ intermediate shape is emphasized. To capture the ultimate shape of the object, $M_U$, the complete sequence $S_{seq}$ would be directly rendered without hiding any entities. The rendering process can be found in the "CAD Code Refinement" of Figure 1. Finally, each $I_0$ can be defined with a set of perspective images $M$ consisting of $M_I$ and $M_U$:

$$M_I = [R(S_1), R(\bar{S}_1 \oplus S_2), \cdots, R(\bar{S}_1 \oplus \bar{S}_2 \oplus \cdots \oplus S_n)], \tag{4}$$

$$M_U = R(S_1 \oplus S_2 \oplus \cdots \oplus S_n), \tag{5}$$

$$M = [M_I, M_U], \tag{6}$$

where $\oplus$ represents the concatenation of SSR triplets. In practice, only a single image is rendered for each $S_K$. Benefiting from the highlight of each entity in the step-wise images, this effectively avoids occlusion issues encountered in the single-view rendering.

**(2) Query for Calling Feedback.** Instead of using VLMs to judge the alignment between the initial description $T$ and $M$, Seek-CAD directly make it judge the alignment between the thought from DeepSeek-R1, $CoT = [t_1, t_2, \ldots, t_m]$, and $M$. $t_i \in CoT$ depicts a clear design logic of the current step according to the initial description $T$, which is highly compatible with current step-wise image $S_K$. This would help VLMs understand clearly how the object is constructed. Next, we feed $M$ along with its corresponding thought $CoT$ into Gemini (VLM) and prompt it to judge the alignment between $M$ and $CoT$ to generate feedback $F_{call}$, this process can be defined as:

$$F_{call} \sim P(F_{call} \mid G, M, CoT), \tag{7}$$

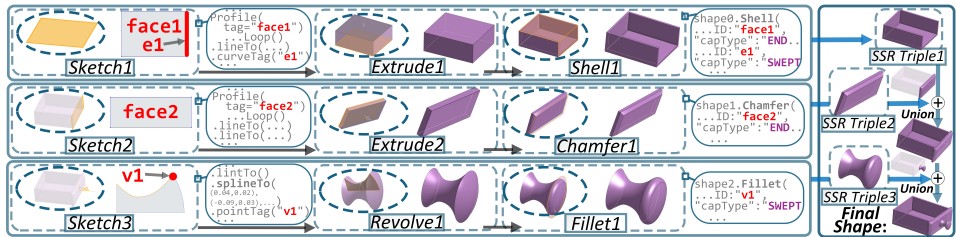

Figure 2: The SSR Design Paradigm. Each CAD model is constructed as a sequence of SSR triplets, where each triplet consists of a *sketch*, a sketch-based feature (e.g., *extrude*, *revolve*), and optional refinement features (e.g., *shell*, *chamfer*, *fillet*). Topological primitives is traced using the *CapType* reference system (*START*, *SWEPT*, *END*) during modeling operations. Final shapes are formed by applying boolean operations (e.g., Union, Cut, Intersect) between the outputs of SSR triplets.

where $G$ is our prompt query designed with two rules for generating $F_{call}$: (i) If $M$ and *CoT* are well aligned, $F_{call}$ should be a positive feedback to return. (ii) If $M$ and *CoT* are mismatched, $F_{call}$ would outline a clear statement to point out discrepancies between $M$ and *CoT*, which would be delivered back to DeepSeek-R1 to again refine the generated CAD code (Recall Equation 3).

## 4 SSR TRIPLE DESIGN PARADIGM

**(1) SSR Triple Design definition.** The Sketch and Extrude (SE) paradigm persists as the predominant approach in contemporary feature-based CAD modeling Khan et al. (2024a); Xu et al. (2023); Zhang et al. (2025b); Wang et al. (2024); Yavartanoo et al. (2024), attributed to its versatility and facilitation of parametric editing. Current large-scale datasets for parametric sequence modeling Wu et al. (2021); Willis et al. (2021) also adopt this paradigm. Nevertheless, the command sets provided by preceding datasets are constrained to basic operations such as *sketch* and *extrude*, and the curves incorporated within sketches are predominantly confined to elementary types. Consequently, contemporary research endeavours leveraging these datasets tend to yield simpler and less diverse geometric forms, failing to adequately address the complexities inherent in real-world design requirements. To address this, we introduce a novel modeling paradigm called SSR (Sketch, Sketch-based feature, and Refinements), in which each modeling step is represented as a SSR triplet $S$:

$$S = (s, f, \langle r_1, r_2, \ldots, r_k \rangle \text{ or } \varnothing), \tag{8}$$

where $k \geq 0$, $s$ denotes a 2D *sketch* feature, $f \in \mathcal{F}$ is a sketch-based feature such as *extrude* or *revolve*, and $\langle r_1, r_2, \ldots, r_k \rangle$ is an ordered sequence of zero or more refinement features, where each $r_i \in \mathcal{R}$ and $\mathcal{R}$ includes features such as *fillet* and *chamfer*. Each SSR triplet $S_i$ is compiled into a 3D geometry $\mathcal{S}_i = \text{CAD\_Kernel}(S_i)$ via a CAD kernel like PythonOCC, and a complete CAD model $\mathcal{M}$, composed of $n$ SSR triples, is represented as a sequence of geometry units joined by boolean operations:

$$\mathcal{M} = \langle \mathcal{S}_1, \text{op}_1, \mathcal{S}_2, \text{op}_2, \ldots, \mathcal{S}_n \rangle, \tag{9}$$

where $\text{op}_i \in \{\text{Union}, \text{Cut}, \text{Intersect}\}$ denotes a boolean operation applied between adjacent geometry units. An example of this process is shown in Figure 2.

**(2) *CapType* Reference Mechanism.** In the context of complex geometric modeling, the integration of refinement features requires referencing particular topological primitives generated during intermediate modeling phases, which are not retained within the design history (parametric model). Consequently, we introduce the *CapType* reference mechanism to address this issue.

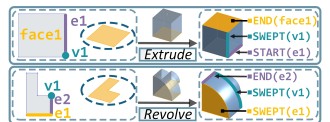

Figure 3: Illustration of the proposed *CapType* reference mechanism.

Given an SSR triplet $S = (s, f, \langle r_1, r_2, \ldots, r_k \rangle)$, where $s$ is the 2D *sketch* containing a set of primitives $\mathcal{A} = \{a_1, a_2, \ldots, a_m\}$, we define the intermediate geometry $\mathcal{S}' = \text{CAD\_Kernel}(S')$, where $S' = (s, f)$ omits the refinement operations. The resulting geometry $\mathcal{S}'$ contains a set of primitives $\mathcal{B} = \{b_1, b_2, \ldots, b_n\}$. The *CapType* reference mechanism defines a mapping $\mathcal{A} \to \mathcal{B}$ as:

$$\phi(a, C) \to b, \quad a \in \mathcal{A}, \quad b \in \mathcal{B}, \quad C \in \{\text{START}, \text{END}, \text{SWEPT}\}, \tag{10}$$

where $C$ denotes the *CapType* category: *START* and *END* refer to the primitives at the start and end of the 3D operation $f$, respectively, while *SWEPT* refers to the primitives generated along the trajectory of the operation, as illustrated in Figure 3. This mechanism allows each refinement operation $r_i \in \mathcal{R}$ to reference a specific primitive $b \in \mathcal{B}$ via $\phi(a, C)$, enabling precise and reliable identification. As illustrated in Figure 5(c) and Figure 7, the method enables the generation of complex and diverse CAD models that align well with real-world design requirements.

## 5 EXPERIMENTS

### 5.1 EXPERIMENTAL SETTINGS

**(1) Metrics.** To evaluate how precisely the generated CAD commands depict the 3D object, we sample 2000 points separately from CAD models in Ground Truth *GT* and generated CAD models *D*. We use **Chamfer Distance (CD)**, **Hausdorff Distance (HD)**, and **Intersection over the Ground Truth (IoGT)** Alrashedy et al. (2025) to measure the differences between *GT* and *D*. Besides, we utilize **G-Score** (score from 1 to 5, allowing decimals by Gemini-2.0) to judge the alignment between the description and the perspective image of its corresponding generated CAD model. To prove that the generative model is not a mere replication of the local CAD corpus, we also compute a **Novel** metric as: $\frac{1}{n}\sum_{i=1}^{n}\mathbb{I}[s(I_A, I_{B_i}) < \tau] \geq \rho$, where $I_A$ is a rendering image of the generated CAD model by SeekCAD and $I_{B_i}$ is a rendering image of each CAD model in the local CAD corpus, and $s(*)$ is a similarity function ($\tau = 0.8$, $\rho = 0.8$). In practice, we use ResNet-50 He et al. (2016) to encode $I_A$ and $I_{B_i}$ before calculating their similarity. Note that cases failed to compile would be excluded when calculating these metrics. Given the generated CAD commands could be failed to compile, we further add **Pass@k** to better understand the mechanism of components in Seek-CAD. **Pass@k** denotes for each description, the probability of at least one set of generated CAD commands being compiled successfully in $k$ times generation. We report mean values under all metrics.

**(2) Comparison Methods.** We have witnessed two training-free efforts of generative CAD modeling with the self-refined strategy: 3D-PreMise Yuan et al. (2024) and CADCodeVerify Alrashedy et al. (2025). To compare with them fairly, only the refined strategy in Seek-CAD (SVF) is separately replaced with refined strategies in 3D-PreMise and CADCodeVerify, while the rest of the components in Seek-CAD are all consistent. Besides, we choose the latest finetune strategy (CAD-Llama Li et al. (2025b)) as a competitor to show the capability of Seek-CAD.

**(3) Implementation Details.** We deploy DeepSeek-R1:32B in Q4 quantization version as a backbone of Seek-CAD on one NVIDIA RTX 3090 GPU with Ollama Ollama. The context length is set to 15,000 to endow inference speed of 21.78 tokens per second. For the inference stage, we set temperature $T = 0.7$, top-p as 0.8 to make it capable of generating diverse CAD models in different trials. For RAG settings, we employ Dify Dify and bge-m3 Chen et al. (2024), details refer to Appendix A.4. For the refinement stage, we adopt the Gemini-2.0 API Team et al. (2024) from Google AI to judge the discrepancies between step-wise perspective images and their corresponding thoughts (DeepSeek-R1:32B-Q4). The feedback from Gemini would be delivered back to Seek-CAD to again refine the generated CAD commands. We set the maximize iteration step of 1 for the refinement stage.

### 5.2 EXPERIMENTAL RESULTS

**(1) Generation.** To avoid data overlapping, we construct a test set consisting of an additional 500 CAD models which are completely different from 10, 000 CAD models in the local CAD corpus. Based on our framework, we compare our SVF with two other visual feedback strategies in one-step refinement. As shown in Table 1, the SVF strategy employed in Seek-CAD significantly outperforms the feedback methods used in 3D-Premise and CADCodeVerify across all metrics. This demonstrates that the CoT and step-wise images utilized in the SVF strategy are indeed well-aligned, and the object design logic contained within the CoT empowers the VLM (Gemini-2.0) to provide clearer feedback, enabling the

Table 1: The quantitative results of generation ability tested on 500 CAD models.

| Strategy | Method | CD↓ | HD↓ | IoGT↑ | G-Score↑ | Novel↑ |
|---|---|---|---|---|---|---|
| **Finetune** | CAD-Llama | 0.2147 | 0.5864 | 0.7023 | 3.3385 | **77.64%** |
| **Training-free** | 3D-PreMise | 0.2203 | 0.6137 | 0.6315 | 3.2022 | 49.57% |
| | CADCodeVerify | 0.2164 | 0.5917 | 0.6562 | 3.3927 | 55.38% |
| | Seek-CAD (Ours) | **0.1979** | **0.5566** | **0.7226** | **3.5185** | 64.04% |

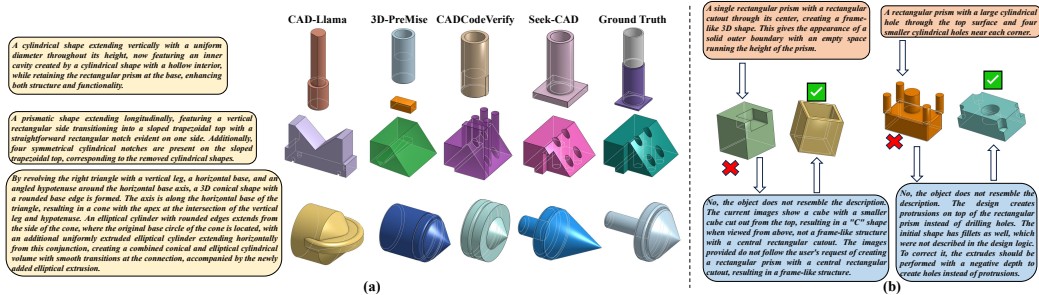

Figure 4: (a) Visual illustrations of CAD generative comparison. (b) The visualizations of refinement capability through the SVF strategy (Recall Sec 3.2). Please enlarge to 225% to see the text clearly.

more precise refinement. Besides, all feedback methods based on our framework achieve close to or over 50% under the Novel metric. This indicates that while the local CAD corpus serves as a constraint on CAD command generation, it does not fully limit the creative ability of our framework to generate novel CAD models. Figure 4(a) gives comparable showcases of Ground Truth *GT* and the generated CAD models from Seek-CAD with different refined strategies. Compared to 3D-PreMise and CADCodeVerify, the SVF embedded in Seek-CAD makes the generated CAD model closer to *GT*, which again proves that SVF (Sec 3.2) is reasonable and effective to integrate step-wise images and the thought from DeepSeek-R1. Besides, Seek-CAD shows superior performance across CD, HD, IoGT, and G-Score metrics compared to CAD-Llama. This highlights its ability to prioritize geometric accuracy and achieve higher text-matching fidelity, proving a stronger semantic understanding. Although it scores lower on novelty than CAD-Llama, this trade-off results in models that are geometrically closer to the ground truth (Figure 4(a)). Crucially, Seek-CAD's ability to completely bypass the training phase shows it can rival the overall performance of fine-tuned models, presenting a significant advantage when training resources are limited.

**(2) Refinement Step.** To better understand the impact of visual feedback on CAD model generations, we explore the effects of performing 0, 1, and 2 rounds of SVF within Seek-CAD, which can be found in Table 2. Compared to Round 0, the generated CAD models show significant improvements across four metrics (CD, HD, IoGT, G-Score) after undergoing corrections in Round 1 and Round 2, which proves the effectiveness of SVF. Additionally, we found that the gains diminish significantly with an increasing round of SVF iterations. For instance, compared to

Table 2: The quantitative results of comparing refinement rounds tested on 500 CAD models.

| Refine Round | Pass@2↑ | CD↓ | HD↓ | IoGT↑ | G-Score↑ |
|---|---|---|---|---|---|
| 0 | 0.77 | 0.2275 | 0.6194 | 0.6183 | 3.1401 |
| 1 | 0.72 | 0.1979 | 0.5566 | 0.7226 | 3.5185 |
| 2 | 0.55 | 0.1966 | 0.5548 | 0.7347 | 3.5314 |

Round 1, Round 2 shows only marginal improvements on the four metrics (e.g., IoGT from 0.7226 to 0.7347), but it increases the probability of code compilation failures (Pass@2 from 0.72 to 0.55). There are two main reasons for this: (i) Limited by the inherent reasoning ability of the base model (DeepSeek-R1:32B-Q4) in Seek-CAD, it is difficult for the model to make perfectly precise corrections based on SVF feedback. (ii) There is a certain probability that the generated CAD commands will fail to compile during each generation process. Each additional round of refinement would increase its probability. Hence, we set the maximum iteration step of 1 for the refinement stage to avoid excessive and potentially unnecessary modifications. Two cases are depicted in Figure 4(b). In the right-hand instance, the initially generated CAD model exhibits a reversed extrusion direction, resulting in the failure to form holes in the corners and the middle of the object, which is resolved after one round of refinement. It again demonstrates the effectiveness of the SVF strategy.

**(3) Ablation Study.** To figure out the mechanism of Seek-CAD, we conduct ablation studies on 200 CAD models by removing some components in Seek-CAD to achieve an additional 7 frameworks tagged from A to G, which separately denote: A (*w/o local CAD corpus*), B (*w/o knowledge constraint*), C (*w/o inter images in SVF*), D (*w/o ultimage image in SVF*), E (*w/o vector search in local CAD corpus*), F (*w/o text search in local CAD corpus*), and G (*w/o CoT in SVF*). For the complete version of Seek-CAD, we denote it as H. Note that all comparisons in the ablation study are based on the one-time refinement. As shown in Table 3, model A completely fails to generate compilable CAD commands, indicating that within a training-free framework, the local

CAD corpus indeed imposes constraints to guide model to generate CAD commands based on the SSR format. This is highly effective for the Seek-CAD model, which lacks prior knowledge of SSR, providing a technical approach for quickly achieving CAD model generations without any training. Compared to model H, the impact of models B, E, and F is primarily focused on the compilability of generated CAD commands. In particular, the substantial decline in Pass@1 for model B (from 0.68 to 0.44) highlights the critical role of the system prompt as an effective constraint during CAD command generation. Besides, models E and F also exhibit a significant decrease in four other metrics (e.g., from 0.7451 to 0.6464 in IoGT, 3.8409 to 3.6315 in G-Score).

This proves conducting hybrid research in the local CAD corpus is more effective. Furthermore, we found that the image utilization strategy within SVF mainly impacts the precision of the refinement, as evidenced by model C achieving worse scores across all metrics (e.g., 0.2295 vs 0.2114 in CD) compared to model D. This validates that inter-images can provide more visual information than the ultimate-image, thereby enabling the VLM to better understand the construction process of objects and generate clear feedback to enhance further modifications. Model G shows a decline in generation quality (e.g., G-Score from 3.8409 to 3.6120) compared to model H. It

Table 3: Ablation Studies on 200 CAD models.

| | Pass@1↑ | Pass@2↑ | CD↓ | HD↓ | IoGT↑ | G-Score↑ |
|---|---|---|---|---|---|---|
| A | - | - | - | - | - | - |
| B | 0.44 | 0.64 | 0.2295 | 0.6307 | 0.6287 | 3.5896 |
| C | 0.67 | 0.79 | 0.2114 | 0.5914 | 0.6713 | 3.7254 |
| D | 0.67 | 0.80 | 0.1961 | 0.5573 | 0.7036 | 3.7761 |
| E | 0.56 | 0.77 | 0.2178 | 0.5947 | 0.6563 | 3.6642 |
| F | 0.47 | 0.68 | 0.2235 | 0.6188 | 0.6464 | 3.6315 |
| G | 0.65 | 0.76 | 0.2247 | 0.6254 | 0.6373 | 3.6120 |
| H | 0.68 | 0.81 | 0.1923 | 0.5382 | 0.7451 | 3.8409 |

indicates CoT helps Gemini-2.0 better understand the step-wise images and improve the quality of feedback. Finally, the complete model H surpasses other comparable versions across all metrics (e.g., highest score 0.7451 in IoGT), proving the design components in Seek-CAD are all effective and reasonable.

**(4) VLM feedback quality.** To provide a deeper insight, we conducted a new statistical analysis of the VLM feedback, categorizing the responses into three types: **"Yes" (Aligned)**,**"No" (Misaligned)**, and**"Unsure"** .We performed this analysis on the 500 CAD samples from our test set (utilizing the optimal refinement step $N = 1$). The detailed statistics are presented in Table 4. Based on it, the impact of each type can be summarized:

(i) "Unsure" (Useless): This category represents cases where the VLM failed to make a definitive judgment. In these cases, the feedback is useless (ineffective), as it does not trigger a meaningful correction, but it effectively acts as a "pass" and does not actively mislead the model. (ii) "Yes" and "No" (Helpful): We carefully verified samples from these two categories by human. We found that these judgments were overwhelmingly valid. "Yes" correctly validated accurate models, and "No" indeed identified

Table 4: The quantitative analysis of the VLM feedback on 500 CAD models.

| Helpful | Useless | Harmful |
|---|---|---|
| ("Yes"/"No") | "Unsure" | - |
| (67/374) | 59 | - |
| 88.2% | 11.8% | - |

logical discrepancies between the CoT and the images. (iii) "Harmful": We acknowledge that harmful hallucinations (e.g., claiming a curve is "not smooth" when it is) are theoretically possible and might appear in larger-scale testing, our manual verification indicates they were not a dominant factor here.

**(5) Performance vary with CAD model complexity.** To further uncover the capabilities of Seek-CAD, we use the total number of CAD commands within the SSR paradigm as the metric for complexity. Specifically, we categorized the generated CAD models into three groups based on their command sequence length: Low [0, 30], Medium [31, 70], and High [71, ). The results can be found in Table 5.

The increased sequence length makes the generation task harder, increasing the difficulty of perfectly matching the target Ground Truth (leading to lower geometric scores). However, it also introduces more degrees of freedom in the generation process. This enhances the diversity of the generated models, thereby boosting their Novelty.

Table 5: The results of performance vary with CAD model complexity.

| Length | CD↓ | HD↓ | IoGT↑ | G-Score↑ | Novel↑ |
|---|---|---|---|---|---|
| [0, 30] | 0.1898 | 0.5131 | 0.7356 | 3.9324 | 56.25% |
| [31, 70] | 0.2001 | 0.5704 | 0.7021 | 3.4935 | 61.76% |
| [71, ) | 0.2093 | 0.5924 | 0.6759 | 3.1133 | 69.34% |

**(6) Robustness.** We conduct a new set of comparative experiments conducted on the DeepCAD dataset Wu et al. (2021) to validate the robustness of our framework. For this experiment, wereplaced our local RAG corpus with 4,000 CAD models collected from the DeepCAD datasetand another 300 CAD models (different from 4,000 CAD models in the local RAG corpus) as the test set. The comparative results can be found in Table 6. It demonstrates that

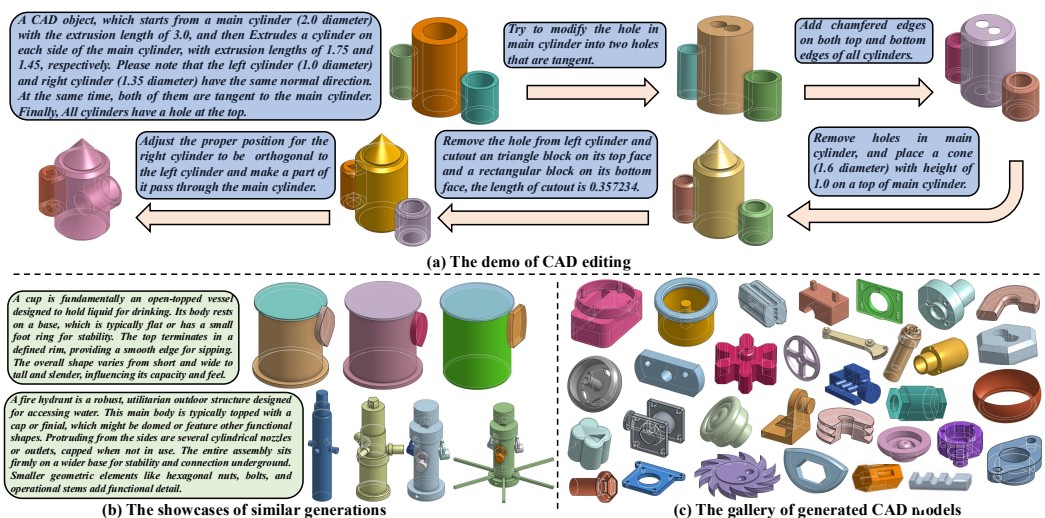

Figure 5: Various Showcases by Seek-CAD. Please enlarge to 180% to see the text clearly.

even on the simpler SE-paradigm dataset, Seek-CAD continues to outperform the baselines across evaluation metrics. This validates the feasibility and robustness of our framework, confirming that its effectiveness is not solely dependent on our proposed SSR-based dataset.

**(7) Various Showcases by Seek-CAD.** We showcase more capabilities of our Seek-CAD, including **CAD editing**, **Similar generations**, and **Gallery of generated CAD models**. **(i) CAD editing** is an applicative sought in a wide range of industrial sectors. As Seek-CAD is a framework built on DeepSeek-R1, it supports multiple rounds of dialogue to achieve substantial editing that includes but is not limited to "Add", "Remove", and "Scale" modification. Figure 5(a) shows a demo to edit the vanilla generated CAD model based on the user's description iteratively. **(ii) Similar generations.** For the same textual descriptions, Seek-CAD can generate similar CAD models as shown in Figure 5(b), which offers useful suggestions for initial CAD design. Besides, the description in Figure 5(b) contains only functional specifications of the objects, in contrast to that in Figure 5(a) which includes specific dimensional parameters. It indicates Seek-CAD is capable of interpreting functional descriptions to produce valid

Table 6: The quantitative results of generation ability tested on the DeepCAD dataset.

| Method | CD↓ | HD↓ | IoGT↑ | G-Score↑ | Novel↑ |
|---|---|---|---|---|---|
| 3D-PreMise | 0.2079 | 0.5823 | 0.7375 | 3.3192 | 48.89% |
| CADCodeVerify | 0.2035 | 0.5745 | 0.7531 | 3.6452 | 51.18% |
| Seek-CAD(ours) | **0.1811** | **0.5231** | **0.8095** | **4.0604** | **54.78%** |

CAD models without relying on explicit parameter guidance. **(iii) The gallery of generated CAD models.** Figure 5(c) provides some generated CAD models from Seek-CAD with feeding textual descriptions from the test set. This shows that Seek-CAD is capable of generating relatively complex models, which was not demonstrated by previous methods based on SE paradigms or CadQuery.

**(8) Limitations and Discussions.** Seek-CAD has several limitations. First, the step-wise visual feedback contains biases from VLMs due to two main issues: (i) VLMs, confined to their training, struggle to accurately describe object geometries without specific-domain training, and (ii) complex models can't be easily summarized. Second, using DeepSeek-R1 for visual feedback introduces redundant information, as full CoTs are fed to VLMs. Third, DeepSeek-R1-32B-Q4 exhibits weaker computational capabilities for geometric constraints compared to DeepSeek-R1-671B model. Specifically, even when the VLM provides correct feedback, the 32B-Q4 model sometimes struggles to generate the precise parameters required to satisfy these constraints. This limitation could be significantly mitigated when scaling up to DeepSeek-R1-671B. Fourth, the *CapType* system struggles with selecting certain topological primitives, like intersection-generated edges, which need intermediate B-Rep modeling data. Specifically, if a logical error is related to a primitive that CapType cannot reference (e.g., "the fillet on that intersection is missing"), the loop cannot currently fix it because the system has no mechanism to point to that specific location.

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

# A APPENDIX

## A.1 KNOWLEDGE CONSTRAINT

We design a knowledge constraint as the system prompt to guide Seek-CAD in generating CAD code that conforms to the SSR paradigm, which helps reduce hallucination to some extent. It consists of three parts including *State of functionality*, *CAD Code Documentation*, and an *Example* as following:

---

**#Statement of functionality#**
*You are a role of CAD-assitant to help user create CAD models. You will be given a description of the shape geometry of 3D objects (e.g., triangle, square, rectangle, and prism), and may contain key parameters (e.g., length, height, width, and radius) and some constraints on the entities (e.g., tangent, parallel, and orthogonal). Your job is to help users complete a whole CAD design logic and show the summary of total CAD commands based on their descriptions. You must follow the rule and definition of each CAD command and its parameter from the #CAD Code Documentation# when you give the total CAD commands code. I will give you a # CAD Code d# to show how it works.*
*#Tips_1# Please give specific and reasonable parameter values to ensure the geometric constraints (e.g., Tangent, Intersect, Orthogonal) between entities when there are multiple sketches.*
*#Tips_2# You cannot ignore any parameter filled with corresponding CAD commands*
*#Tips_3# Finally, please use the most effective and simple CAD command to answer. For example, when the user make a query to construct a chamfer edge, please use "Chamfer" command directly instead of using other CAD commands to replace it.*

**#CAD Code Documentation#**
*### Methods: - `moveTo(x, y)`: Set the starting point of the loop. - `lineTo(x, y)`: Draw a straight line from the current point to `(x, y)`.*
*- `threePointArc(p1, p2)`: Draw a circular arc from the current point through `p1` to `p2`.*
*- `circle(radius)`: Draw a full circle centered at the current point with the given `radius`.*
*- `splineTo(*points)`: Create a smooth spline curve through the given sequence of points from the current point.*
*- `pointTag(tag)`: Assign a tag to the current point (used after curve method).*
*- `curveTag(tag)`: Assign a tag to the most recent curve (used after curve method).*
*## Profile Class: Represents a closed shape made up of one or more loops.*
*### Constructor: - `Profile(tag=None)`: Optionally provide a `tag` to identify the profile.*
*### Method: - `addLoop(*loops)`: Add one or more `Loop` objects to the profile.*
*## Sketch Class: Defines a 2D sketch on a given plane.*
*### Constructor:- `Sketch(plane)`: Must provide a `plane` dictionary with:*
*  - `normal`: A 3D vector `[x, y, z]` indicating the plane's normal direction.*
*  - `origin`: A 3D point `[x, y, z]` specifying the sketch's origin.*
*  - `x`: A 3D vector `[x, y, z]` defining the X-axis direction in the plane.*
*### Method: - `addProfile(*profiles)`: Add one or more `Profile` objects to the sketch.*
*## Shape Creation*
*### `Extrude(sketch, distance=(a, b)) → Shape`: Create a 3D solid by extruding the profiles in a sketch.*
*- `a`: Extrusion distance in the sketch's normal direction. - `b`: Extrusion distance in the opposite direction.*
*### `Revolve(sketch, axis={point, direction}, angle=(a, b)) → Shape`: Create a 3D solid by revolving a profile around a given axis.*
*- `point`: A 3D point `[x, y, z]` on the axis. - `direction`: A 3D vector `[x, y, z]` indicating the axis direction.*
*- `a`: Angle in the normal direction.- `b`: Angle in the opposite direction.*
*## Shape Class: Represents a solid body. Supports Boolean and modification operations.*
*### Boolean Operations: - `shape.union(s: Shape) → Shape`: Merge the current shape with shape `s`.*
*- `shape.cut(s: Shape) → Shape`: Subtract shape `s` from the current shape.*
*- `shape.intersect(s: Shape) → Shape`: Keep only the intersecting volume between the current shape and shape `s`.*
*### Modifiers: - `shape.Chamfer(width, entities)`: Apply a chamfer to tagged edges or curves.*
*- `shape.Fillet(radius, entities)`: Round corners with a fillet radius. - `shape.Shell(thickness, entities)`: Hollow out the shape with specified wall thickness.*

**#Example#**
*# Description # The resulting model is an L-shaped prismatic solid with two perpendicular rectangular arms, one extending vertically and the other horizontally, forming the characteristic L shape in 3D space. Additionally, there are four vertical rectangular pillar-like prisms extending downward from each corner of the virtual bounding rectangle.*
*----*
*# Generated Answers # Based on your description, I can help you create the CAD command summary for generating the described shape. Here is the complete CAD code command design with valid parameters:*
*sk0 = Sketch(plane={"normal": [0.0, 0.0, 1.0], "origin": [0.0, 0.0, 0.0], "x": [1.0, 0.0, 0.0]})*
*p0_0 = Profile()*
*p0_0.addLoop(Loop().moveTo(0.0,-2.667).lineTo(0.254,-2.667).lineTo(0.254,-0.254)*
*    .lineTo(1.8288,-0.254).lineTo(1.8288,0.0)*
*    .lineTo(0.0,0.0)*
*    .lineTo(0.0,-2.667))*
*sk0.addProfile(p0_0)*
*shape0 = Extrude(sk0, distance=0.9144)*
*sk1 = Sketch(plane={"normal": [0.0, 0.0, 1.0], "origin": [0.0, 0.0, 0.0], "x": [1.0, 0.0, 0.0]})*
*p1_0 = Profile()*
*p1_0.addLoop(Loop()*
*...*
*sk1.addProfile(p1_0,p1_1,p1_2,p1_3)*
*shape1 = Extrude(sk1, distance=0.8636)*
*shape0 = shape0.union(shape1)*

---

Figure 6: The knowledge constraint adopted in Seek-CAD.

## A.2 CAD DATASET BASED ON SSR PARADIGM

We extend the data parsing methodology of DeepCAD by utilizing Onshape's Developer API and FeatureScript to parse CAD models based on the ABC dataset. Our method supports core modeling commands including *sketch*, *extrude*, *revolve*, *fillet*, *chamfer*, and *shell*. During the parsing process, models containing unsupported operations (such as mirror) are not immediately discarded. Instead, we filter them only if the proportion of unsupported commands exceeds a threshold of 0.2. Unsupported commands are ignored, and only supported operations are retained for further processing.

For commands like fillet, chamfer, and shell, we first extract the primitives they operate on (e.g., edges for *chamfer* or faces for *shell*), and then use PythonOCC APIs to match these primitives to specific sketch elements through their *CapType* designation. We note that the proposed *CapType* reference system cannot identify certain edges or faces that are not directly associated with sketch-defined primitives such as those generated by the intersection of solid bodies. Therefore, when refinement commands (*chamfer*, *fillet* or *shell*) involve such primitives that cannot be identified via *CapType* schema, we exclude those primitives from the commands. Finally, all parsed data is converted into a unified JSON format compatible with DeepCAD, which can be directly rendered using PythonOCC for visualization.

To further illustrate the characteristics of our dataset, Figure 7 presents a qualitative comparison between our dataset and DeepCAD. The statistical distributions of command sequence lengths and the number of curves per CAD model are shown in Figure 8.

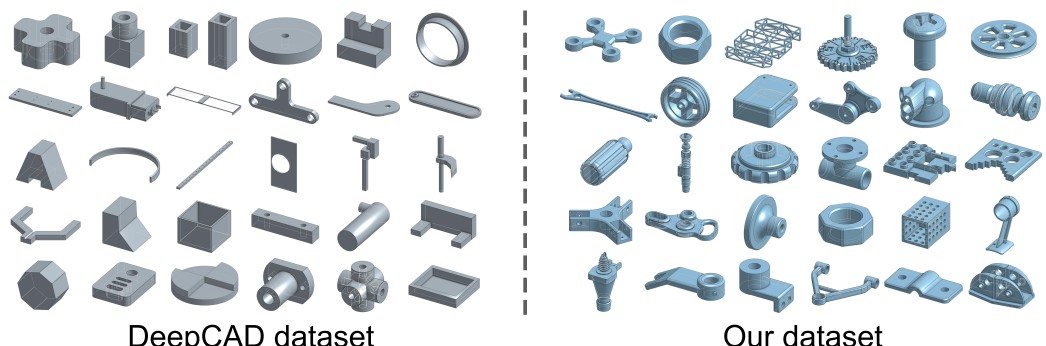

DeepCAD dataset                Our dataset

Figure 7: Qualitative comparison between our dataset and the DeepCAD Wu et al. (2021) dataset. Compared to DeepCAD, our dataset captures more realistic and structurally complex industrial designs, supporting a broader range of modeling operations such as spline, revolve, chamfer, fillet, and shell. Our extended dataset also features richer geometric details and greater diversity in modeling strategies.

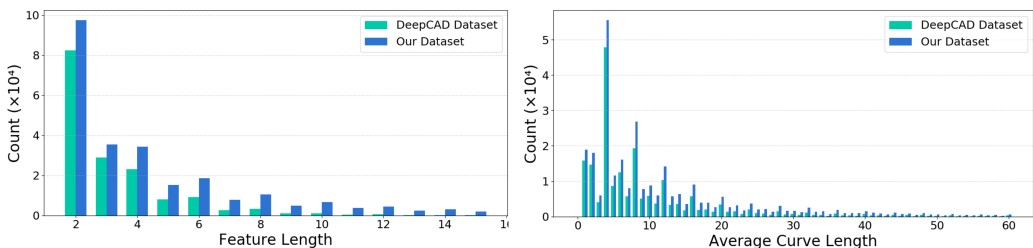

Figure 8: Statistical comparison between our complete dataset and the DeepCAD Wu et al. (2021) dataset, which also is a subset of our dataset. The left plot illustrates the distribution of feature lengths, while the right plot shows the distribution of average curve lengths.

### A.3 CAD CODE REPRESENTATION

Since existing CAD scripting tools such as CADQuery do not directly support *CapType* reference mechanism and SSR design paradigm, we define a concise and effective Python-based representation tailored for SSR-driven CAD modeling. As shown in Listing 1, the CADShape class encapsulates an SSR triplet and supports boolean operations—such as union, cut, and intersect—with other SSR triplets. The final CAD model is represented by a CADShape instance, which can either be a single SSR triplet or the result of boolean operations among multiple SSR triplets.

For refinement features (e.g., chamfer, fillet, and shell), the target edge or face primitives are determined using the *CapType* introduced in Section 4. In particular, applying a chamfer or fillet to a face affects all edges on that face. Finally, we convert the code into the JSON format similar to DeepCAD, which is then rendered using PythonOCC.

Listing 1: Python interface for SSR-based CAD modeling. The CADShape class provides a concise and extensible representation that supports structured modeling under the SSR design paradigm.

```python
class Loop:
    def __init__(self) -> None: ...
    # Define start point of this loop
    def moveTo(self, x: float, y: float) -> Self: ...
    # Draw straight line to point
    def lineTo(self, x: float, y: float) -> Self: ...
    # Arc to p2 via p1
    def threePointArc(self, p1: Tuple[float, float], p2: Tuple[float,
                                               float]) -> Self: ...
    # Spline through given points
    def splineTo(self, *p: Tuple[float, float]) -> Self: ...
    # Close loop (make a line back to start point)
    def close(self) -> Self: ...
    # Draw circle with center at current point
    def circle(self, radius: float) -> Self: ...
    # assign tag to the current point
    def pointTag(self, tag: str) -> Self: ...
    # assign tag to the current curve
    def curveTag(self, tag: str) -> Self: ...

class Profile:
    # Create a face with a optional tag for reference
    def __init__(self, tag: Optional[str] = None) -> None: ...
    # Add one or more loops to the face
    def addLoop(self, *loops: Loop) -> None: ...

class Sketch:
    # plane format: {"origin": [x, y, z], "x_axis": [x, y, z], "normal":
                                       [x, y, z]}
    def __init__(self, plane: Dict) -> None: ...
    def addProfile(self, *profiles: Profile) -> None: ...

class CADShape(ABC):
    def __init__(self) -> None: ...
    # entities: list of dicts, each with {"capType": "START"|"END"|"SWEEP
                                       ", 'referenceId': str},
                                       specifying referenced entities
    def Chamfer(self, width: float, entities: List[Dict]) -> Self: ...
    def Fillet(self, radius: float, entities: List[Dict]) -> Self: ...
    def Shell(self, thickness: float, entities: List[Dict]) -> Self: ...
    def union(self, shape: CADShape) -> Self: ...
    def cut(self, shape: CADShape) -> Self: ...
    def intersect(self, shape: CADShape) -> Self: ...
```

```python
class Extrude(CADShape):
    def __init__(self, sketch: Sketch, distance: Union[float, Tuple[float
                                      , float]]) -> None: ...

class Revolve(CADShape):
    # axis format: {"point": [x, y, z], "direction": [x, y, z]}, axis is
                                      defined by a point and a
                                      direction
    def __init__(self, sketch: Sketch, axis: Dict, angle: Union[float,
                                      Tuple[float, float]]) -> None:
                                      ...
```

### A.4 RAG Settings

To equip the SSR-based CAD corpus locally for the retrieval-augmented generation of Seek-CAD, we use Dify Dify in 0.15.3 version, which is a docker-based Docker platform. For each query, we use bge-m3 Chen et al. (2024) to embed it into a vector that is used to calculate similarity score with CAD models stored in the local SSR-based corpus. Practically, we select Top-3 samples based on a hybrid searching strategy (Sec 3.1) for the augmentation retrieval. To incorporate the retrieved content into the input for Seek-CAD, we apply a chunk-based prompt construction strategy. Each retrieved pair $(d_j, s_j) \in \mathcal{R}_T$ is formatted as:

$$[\text{Chunk}_j] := \texttt{"Description: "} d_j \, \| \, \texttt{"CAD code: "} s_j, \tag{11}$$

where $\oplus$ denotes string concatenation. The final prompt for Seek-CAD becomes:

$$T \oplus (d_j, s_j) = \texttt{"Query: "} T \oplus \text{Chunk}_1 \oplus \ldots \oplus \text{Chunk}_k. \tag{12}$$

### A.5 Novel Visualizations

Benefiting from the prior knowledge of DeepSeek-R1, Seek-CAD is capable of generating not only industry-oriented CAD models but also non-industrial ones that differ in style from those in our local CAD corpus. As shown in Figure 9, this demonstrates that Seek-CAD does not rely entirely on our local CAD corpus and is able to generate models with a certain degree of novelty.

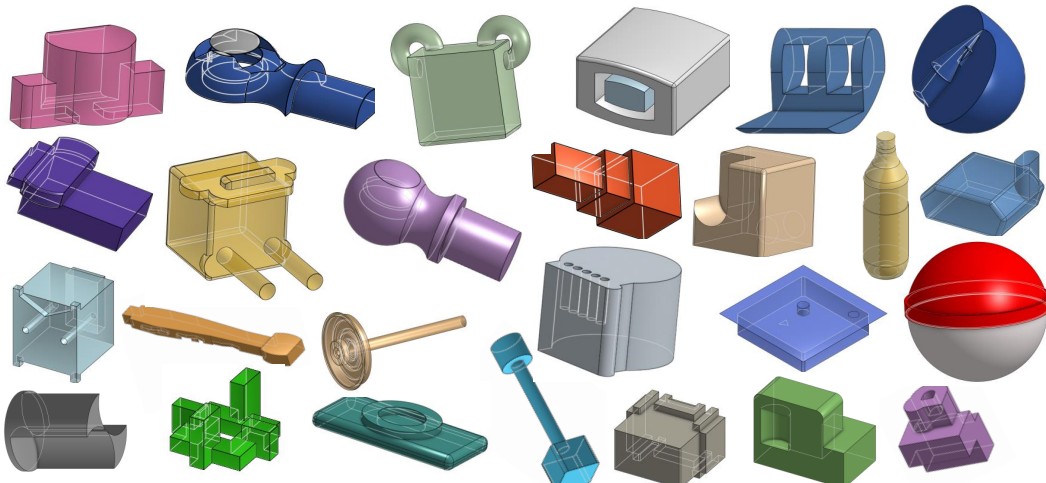

Figure 9: Novel showcases generated by Seek-CAD, which are different from the style of CAD models in the local CAD corpus.

## A.6 ENLARGED VERSION OF SHOWCASES IN FIGURE 5(C) OF THE MAIN MANUSCRIPT

We provide an enlarged version of showcases in Figure 5(c) for a clear view as shown in Figure 10.

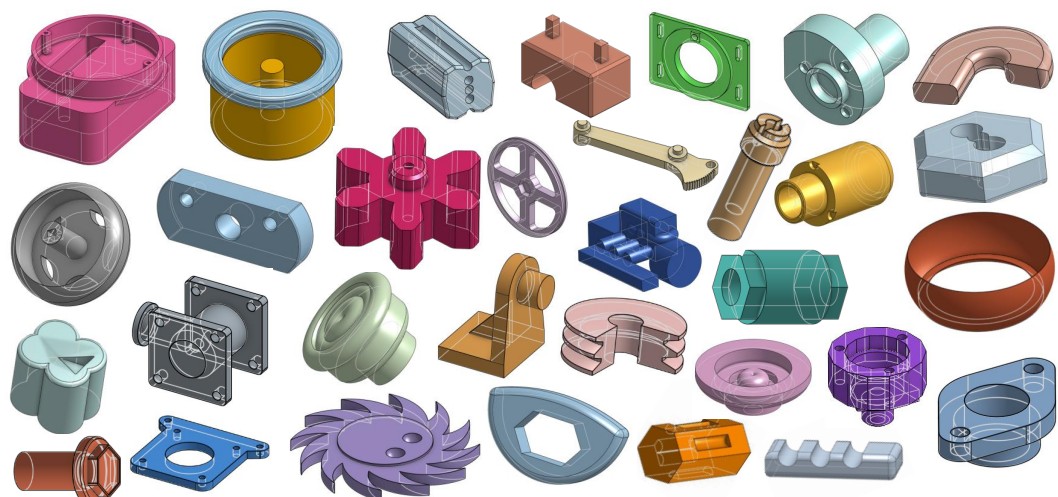

Figure 10: The enlarged version of showcases in Figure 5(c) of the main manuscript.

## A.7 MORE SHOWCASES OF CAD EDITING

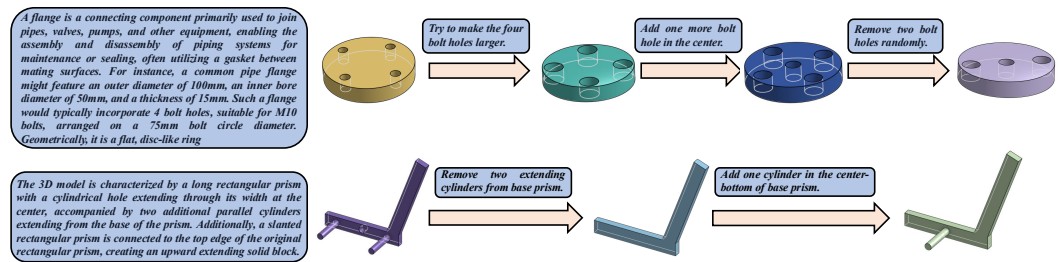

Figure 11: The showcases of similar generations.

## A.8 MORE SHOWCASES OF SIMILAR GENERATIONS

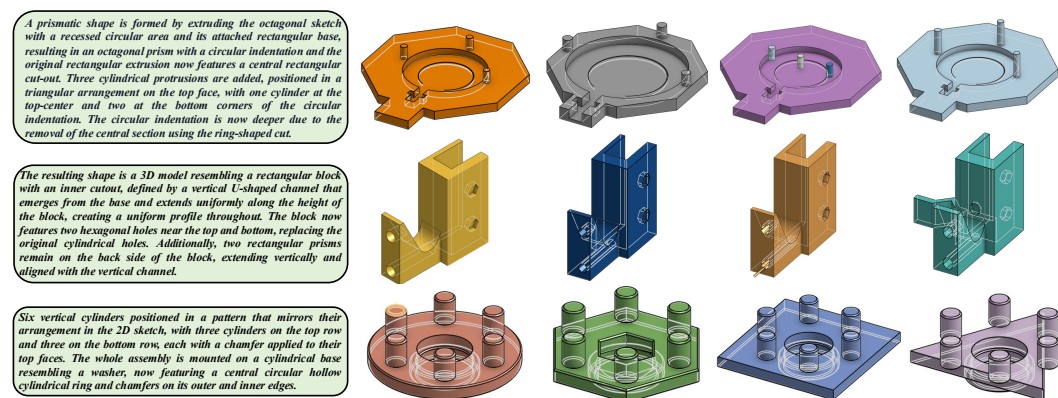

Figure 12: The showcases of similar generations.

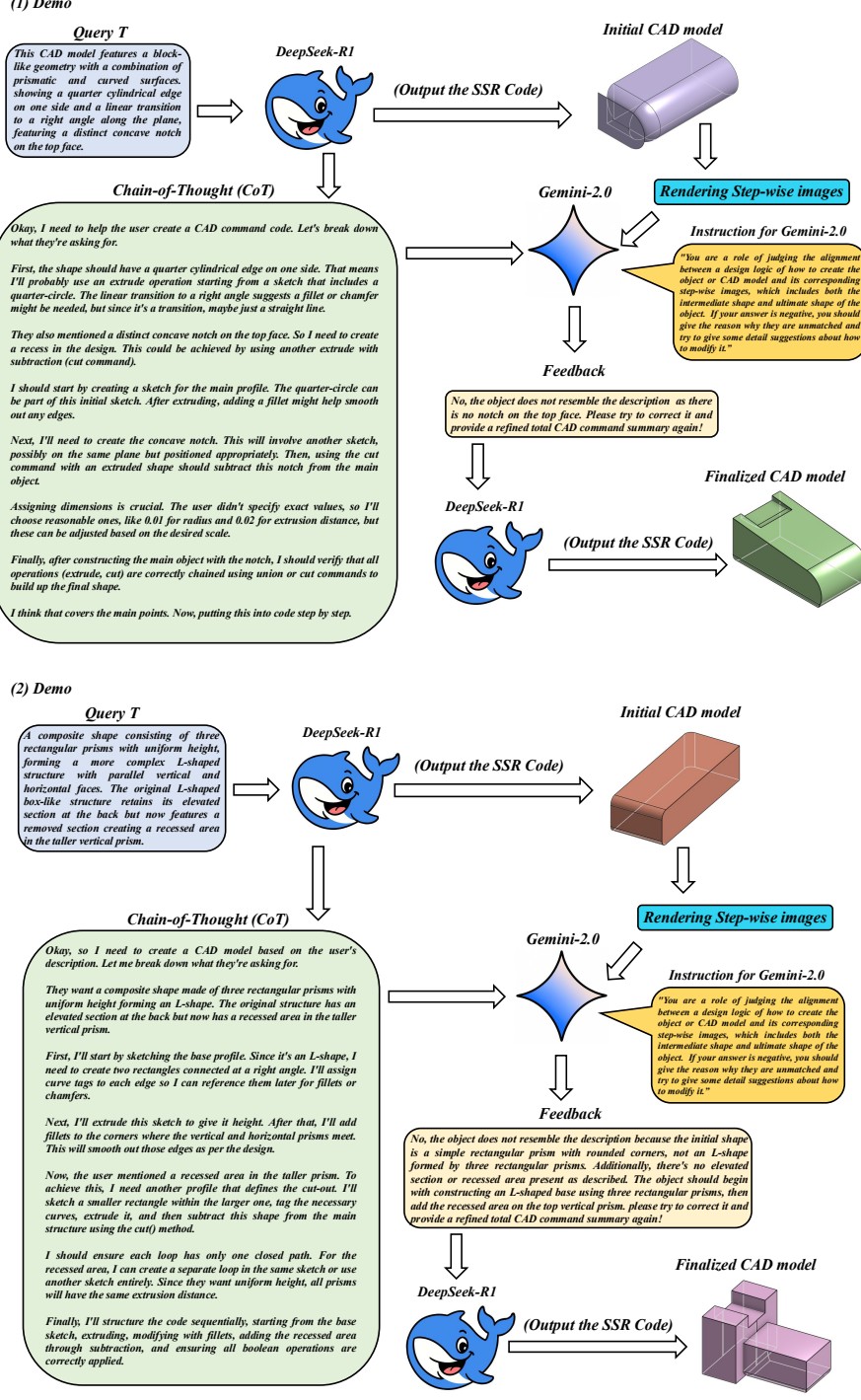

Figure 13: The showcases of complete generation process.

## A.9 THE DETAIL SHOWCASE OF COMPLETE GENERATION PROCESS

Here we provide two demo showcases of complete generation process by Seek-CAD, including the input query $T$, CoT from DeepSeek-R1, initial generated CAD models, feedback content, and finalized CAD model (Figure 13).

## A.10 MORE SHOWCASES BY SEEK-CAD

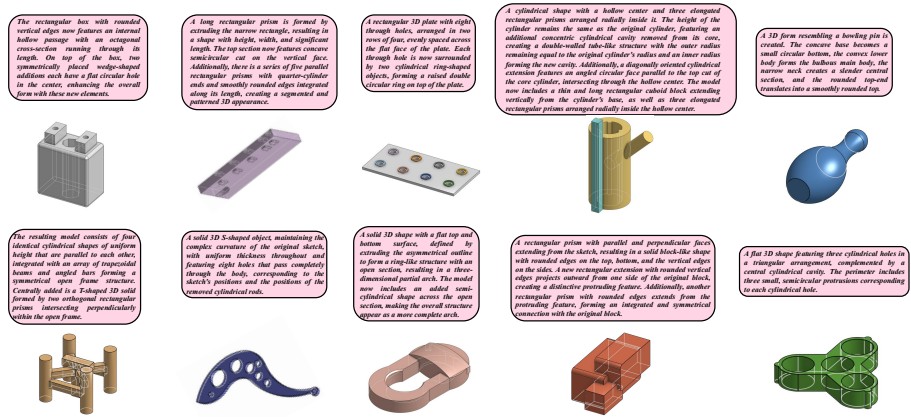

Figure 14: The more generated showcases by Seek-CAD.

## A.11 THE ENLARGED VERSION OF KEY DESIGN MODULES WITHIN SEEK-CAD FRAMEWORK

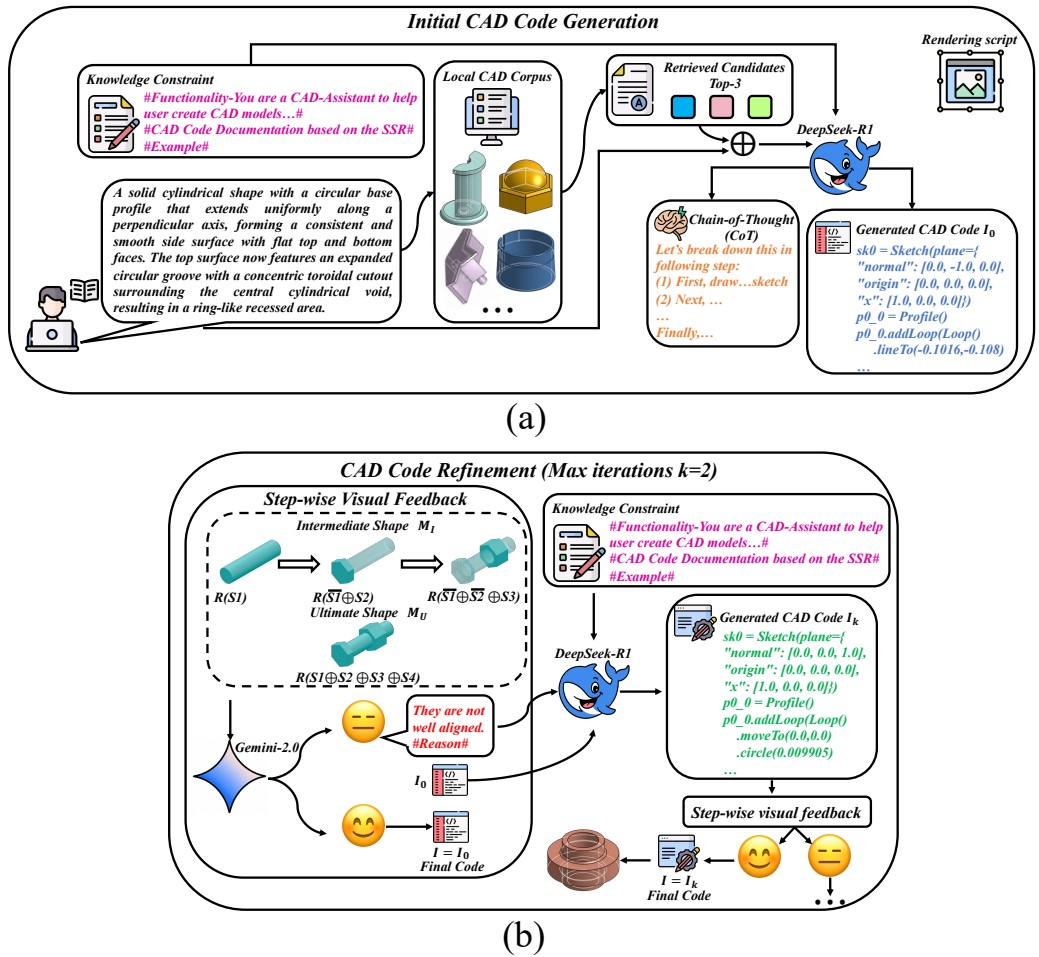

Figure 15: The enlarged version of key design modules: (a) Initial CAD Code Generation. (b) CAD Code Refinement.

