# OpenReview forum: "Seek-CAD: A Self-refined Generative Modeling for 3D Parametric CAD Using Local Inference via DeepSeek"
_ICLR.cc/2026/Conference — ICLR 2026 Poster_

### Official Review · Reviewer_fiYb · 2025-10-30

**Soundness:** 2
**Presentation:** 2
**Contribution:** 2
**Rating:** 4
**Confidence:** 4

**Summary:**

This paper introduces Seek-CAD, a novel framework for generating 3D parametric CAD models. The framework's core features are that it is training-free and locally deployable.

It uses an open-source LLM  as its core, combining RAG with a novel "SSR" design paradigm to generate complex CAD code.

Its core innovation is a self-refinement loop.

Experiments show Seek-CAD exceeds other training-free methods in geometric accuracy and even surpasses the fine-tuned CAD-Llama model.

**Strengths:**

The framework's key innovation is using a VLM to align the LLM's Chain-of-Thought (CoT) with step-wise visual renderings, thereby validating the design process rather than just the final product.
Despite being training-free, Seek-CAD surpasses the fully fine-tuned CAD-Llama on key geometric metrics, proving the potential of its "generate-verify-refine" agent loop.
The system is a practical, self-contained solution, using a locally deployed LLM (DeepSeek-R1) and an essential local RAG corpus instead of relying on expensive, closed-source APIs.

**Weaknesses:**

The entire refinement loop's effectiveness hinges on the VLM's (Gemini-2.0) ability to accurately assess the alignment between the CoT and the step-wise images. The paper admits in its limitations (Sec 5) that VLMs can have biases or misunderstand complex geometry. If the VLM hallucinates or misinterprets (as seen in Fig 4b), it will provide faulty feedback, causing the LLM to make incorrect "corrections."


The "LLM $\rightarrow$ Render $\rightarrow$ VLM $\rightarrow$ LLM" loop is computationally slow. More critically, Table 2 shows that going from Round 1 to Round 2 of refinement yields marginal performance gains (e.g., IoGT only 0.72 $\rightarrow$ 0.73) but causes the code compilation success rate (Pass@2) to drop sharply (from 0.72 to 0.55). This suggests multi-turn refinement may lead the LLM into a state of confusion or over-correction, breaking the code's validity.

**Questions:**

The refinement loop (LLM $\rightarrow$ VLM $\rightarrow$ LLM) is expensive in time, and Table 2 shows its success rate (Pass@k) drops significantly with more iterations. Compared to a one-pass, fine-tuned model (CAD-Llama), is Seek-CAD still competitive in terms of actual wall-clock time and API call costs?


The system relies on a VLM (Gemini) as a "referee" to validate the LLM's (DeepSeek) CoT. If the VLM and the LLM share the same misunderstanding of the user's prompt (e.g., they both misinterpret the meaning of "chamfer"), will the system "confidently" refine toward a wrong answer, because the referee (VLM) will incorrectly validate the executor's (LLM) flawed logic?

---

> ### Author Response · Authors · 2025-11-13
> **Response to Reviewer fiYb Part(1/2)**
>
> We sincerely thank for reviewer’s feedback and constructive criticism. Meanwhile, we are particularly grateful that the reviewer read our paper carefully and recognized the practical potential of our proposed framework in CAD design. Our responses to the reviewer's questions are as follows:
>
> __1. On Weakness 1 & Question 2 ---VLM Reliability and "Shared Misunderstanding"__
>
> We thank the reviewer for this profound question regarding systemic bias. This is a critical challenge. To be honest, we do not deny the possibility of such biases caused by LLM/VLM. Our framework’s core design is precisely intended to mitigate this exact risk. We do not ask VLM to solve the difficult task of judging full description vs. the final shape directly, as this high-level, holistic comparison would increase the likelihood of such biases occurring. This is why we decompose the CAD construction process and render the step-wise images: the explicit goal is to reduce the VLM's task difficulty. As detailed in Sec 3.2, our innovation is to have the VLM assess the alignment between the LLM's Chain-of-Thought (CoT) and the corresponding step-wise rendered images. The CoT provides a clear, step-by-step design logic (e.g., "Now, I will create a chamfer on this edge"). The VLM's task is reduced to a direct verification: "Does the image for this step visually match the logic described in the CoT for this step?" This is fundamentally more robust than the task of full description vs. the final shape validation.
>
> Additionally, to mitigate biases that may arise from the LLM's own misinterpretation, we employ (Retrieval-Augmented Generation) RAG as an auxiliary mechanism. The LLM (DeepSeek) is not operating independently; its generation is strongly constrained by RAG from our 10,000-sample local CAD corpus. When the prompt requests a "chamfer," the RAG retrieves correct code examples of "chamfer" from the corpus. This grounds the LLM's CoT and initial code generation ($I_0$) in factual correctness, which reduces the chance of misunderstandings occurring in the first place.
>
> We directly validated this point in our ablation study. Model G (w/o CoT) represents the exact scenario where the VLM only sees the image and the full description. As shown in Table 3, Model G's performance drops significantly across all metrics (e.g., G-Score from 3.8409 to 3.6120; IoGT from 0.7451 to 0.6373) compared to our full Model H (with CoT). This result empirically proves that our CoT-based feedback is essential and significantly more robust at guiding the VLM than standard visual feedback.
>
> In summary, although it is impossible to completely eliminate the inherent biases in VLMs/LLMs, our (CoT + Step-wise Image) alignment mechanism, combined with RAG, creates a robust system of checks and balances that provides effective control over these potential misunderstandings.

---

> > ### Author Response · Authors · 2025-11-13
> > **Response to Reviewer fiYb Part(2/2)**
> >
> > __2. On Weakness 2 & Question 1: This suggests multi-turn refinement may lead the LLM into a state of confusion or over-correction, breaking the code's validity. The refinement loop (LLM VLM LLM) is expensive in time, and Table 2 shows its success rate (Pass@k) drops significantly with more iterations. Compared to a one-pass, fine-tuned model (CAD-Llama), is Seek-CAD still competitive in terms of actual wall-clock time and API call costs?__
> >
> > We thank the reviewer for this keen and accurate analysis of Table 2. The reviewer's observation is entirely correct. The explicit purpose of the experiment in Table 2 was to determine the optimal number of refinement iterations, balancing performance gains against stability. This is not a failure of our method but rather a key finding that explicitly justifies our final model design. It shows that going from Round 0 to Round 1 provides a massive performance leap across all metrics (e.g., IoGT improves from 0.6183 to 0.7226; G-Score from 3.1401 to 3.5185). It also shows exactly what the reviewer noted: going from Round 1 to Round 2 yields diminishing returns (IoGT +0.0121) while incurring a severe stability cost (Pass@2 drops from 0.72 to 0.55). This finding guides us to set N=1 (a single refinement round) as the optimal strategy. The superior performance of Seek-CAD reported in our main results (Table 1) and ablation studies (Table 3) was achieved using this optimal N=1 strategy. We will clarify this more explicitly in the final manuscript.
> >
> > Besides, about cost competitiveness vs. CAD-Llama. We thank the reviewer raise a fair point about cost. CAD-Llama (a training-based method) and Seek-CAD (a training-free-based method) each have their distinct advantages. To provide a clearer comparison, please allow us to distinguish the 'cost' into two categories: (1) Training Cost and (2) Inference & API Cost.
> >
> > __Training Cost:__ CAD-Llama is a fine-tuned model. It requires a massive training cost (data collection, extensive GPU-days). Seek-CAD is training-free. Its training cost is zero.
> >
> > __Inference & API Cost:__ Although Seek-CAD does consume more inference time compared to CAD-Llama. Seek-CAD trades this zero-training cost for a modest inference cost (the N=1 refinement loop) and API cost: (1) Our core LLM (DeepSeek-R1-32B-Q4) is deployed locally (single RTX 3090 GPU) and is free (0 API cost). (2) The API call to Gemini-2.0 for the one-round refinement (0.1$ for 1M tokens).
> >
> > In conclusion, training-based (CAD-Llama) and training-free (Seek-CAD) represent two different technical paths. The training-based path (CAD-Llama), which relies on data fitting, benefits from faster inference speed and better ensures the validity of the generated models, but this comes at the cost of significant training resources and a potential loss of the base model's (e.g., Llama3) general capabilities due to the supervised fine-tuning. Seek-CAD can entirely bypass this massive cost of fine-tuning and still achieve highly competitive performance with CAD-Llama (e.g., our 0.7226 IoGT vs. 0.7023; our 0.1979 CD vs. 0.2147 in Table 1) by paying the Inference & API Cost. This is a significant advantage in the resource-constrained environment.

---

### Official Review · Reviewer_AAaN · 2025-10-31

**Soundness:** 2
**Presentation:** 3
**Contribution:** 2
**Rating:** 4
**Confidence:** 4

**Summary:**

The paper proposes Seek-CAD, a training-free framework that locally runs DeepSeek-R1-32B-Q4 with RAG over a CAD-code corpus to generate SSR-style CAD programs, then iteratively self-refines them using step-wise visual feedback: rendering intermediate + final shapes and asking a VLM (Gemini-2.0) to judge alignment with the LLM’s CoT, feeding that back for correction.

**Strengths:**

1- Method novelty in feedback: evaluates intermediate renders + CoT rather than final-image only; ablations show inter-image cues matter.

2- Empirical signal: Seek-CAD beats prior training-free refiners and edges a tuned model on geometric fidelity (CD/HD/IoGT), with qualitative evidence.

**Weaknesses:**

1- VLM Feedback Quality: The authors acknowledge that VLMs struggle with geometric descriptions without domain-specific training (Section 5.5), but this fundamental limitation undermines the core refinement mechanism. No quantitative analysis is provided on how often Gemini-2.0's feedback is actually helpful vs. harmful.

2- Compilation Failure Rate: The Pass@k metric reveals concerning compilation failure rates. Even after 2 refinement rounds, only 55% of generated models compile successfully (Table 2). This significantly limits practical applicability. The paper doesn't adequately address strategies to improve this.

**Questions:**

1- Can you provide analysis on when/why Gemini-2.0 feedback helps vs. hurts?

2- Can you provide failure case analysis beyond the two examples in Figure 4(b)?

3- How does performance vary with CAD model complexity (e.g., number of features)?

---

> ### Author Response · Authors · 2025-11-23
> **Response to Reviewer AAaN Part(1/2)**
>
> We sincerely appreciate the reviewer’s valuable feedback. We are encouraged that you highlighted the methodological novelty of our feedback mechanism (SVF + CoT). Your insightful questions have helped us further explore the underlying reasons for the effectiveness of our proposed method. Thanks again! Below are our responses to these questions.
>
> __1. On Weakness 2 about Compilation Failure Rate.__
>
> We thank the reviewer for this keen and accurate analysis of Table 2. We acknowledge that our initial discussion in this section may not have been explicit enough regarding the selection of __N=1__ as the optimal setting, potentially causing some confusion.
>
> The explicit purpose of the experiment in Table 2 was to determine the optimal number of refinement iterations by balancing performance gains against stability. The results explicitly demonstrate that infinite or multiple rounds of refinement are unnecessary and can be detrimental. Based on the Table 2, Based on Table 2, Round 1 demonstrates a superior advantage compared to Round 2, effectively balancing the geometric precision of the generated models with the system's stability (Pass@k).
>
> This key finding (As we stated “Additionally, we found that the gains diminish significantly with an increasing round of SVF iterations.” in the experiment of Refinement Step) guides us to set __N=1 (a single refinement round) as the optimal step__. The superior performance of Seek-CAD reported in our Main Results (Table 1) and Ablation Studies (Table 3) was achieved using this optimal N=1 strategy (where the success rate is a competitive ~72%).
> We will revise the manuscript to explicitly highlight this finding and clarify that Table 2 serves as a justification for stopping at N=1, rather than a proposal to use N=2 in practice.
>
>
>
>
> __2. On Weakness 1 and Question 1 about the analysis of VLM feedback quality.__
>
> We thank the reviewer for raising this crucial question regarding the reliability of VLM feedback. This allows us to clarify the mechanism's practical impact.
>
> __Why Gemini-2.0 feedback helps?__ The most direct quantitative evidence is found in Table 2. The substantial leap from Round 0 (IoGT 0.6183) to Round 1 (IoGT 0.7226) proves that the feedback is overwhelmingly helpful overall. If the feedback were significant harmful, we would not observe such a consistent improvement across all metrics.
>
> __New quantitative analysis:__ To provide a deeper insight, we conducted a new statistical analysis of the VLM feedback, categorizing the responses into three types: __"Yes" (Aligned)__, __"No" (Misaligned)__, and __"Unsure"__. We performed this analysis on the 500 CAD samples from our test set (utilizing the optimal refinement step __N=1__). The detailed statistics are presented in the table below:
> | Helpful    | Useless  | Harmful |
> |------------------------|----------|---------|
> | ("Yes" /"No")  | "Unsure" | -       |
> | (67 / 374)   | 59       | -       |
> | **88.2%**       | **11.8%**| -       |
>
> Based on these statistics, we analyzed the impact of each type:
>
> __"Unsure" (Useless):__ This category represents cases where the VLM failed to make a definitive judgment. In these instances, the feedback is useless (ineffective), as it does not trigger a meaningful correction, but it effectively acts as a "pass" and does not actively mislead the model. (59/500=11.8%)
>
> __"Yes" and "No" (Helpful):__ We carefully verified samples from these two categories by human. We found that these judgments were overwhelmingly valid. "Yes" correctly validated accurate models, and "No" correctly identified logical discrepancies between the CoT and the images. ((67 + 374)/500=88.2%)
>
> __"Harmful":__ Based on this analysis, the VLM feedback in our current experiments primarily acts as either helpful or useless. While we acknowledge that harmful hallucinations (e.g., claiming a curve is "not smooth" when it is) are theoretically possible and might appear in larger-scale testing, our manual verification indicates they were not a dominant factor here.
>
> We will include above analysis in the revised manuscript to provide a clearer picture of the VLM's behavior.

---

> > ### Author Response · Authors · 2025-11-23
> > **Response to Reviewer AAaN Part(2/2)**
> >
> > __3. On question 2 about Can you provide failure case analysis beyond the two examples in Figure 4(b)?__
> >
> > Yes. Beyond the examples in the paper, our analysis identifies two common failure types:
> >
> > __CapType Topological Failures:__ The refinement loop fails when the VLM correctly points out a missing feature (e.g., "missing fillet on the intersection"), but our CapType mechanism cannot reference that specific intersection edge because it wasn't explicitly defined in the original sketch primitives. This leads to un-correctable logical errors.
> >
> > __Constraint Precision Errors:__ The loop sometimes fails to correct subtle "tangency" or "orthogonality" errors. As we state in our limitations, this is largely due to the limited reasoning precision of the smaller DeepSeek-R1-32B model. Even if the VLM provides correct feedback, the 32B model sometimes struggles to generate the precise parameters to satisfy these constraints.  Our preliminary tests show that using the larger 671B model significantly reduces these failures
> >
> > __4. On question 3 about How does performance vary with CAD model complexity?__
> >
> > We thank the reviewer for this insightful question, which helps to further uncover the capabilities of Seek-CAD. To address this, we use the total number of CAD commands within the SSR paradigm as the metric for complexity. We categorized the generated CAD models into three groups based on their command sequence length: Low [0, 30], Medium [31, 70], and High [71, ~). The statistical results are presented in the table below:
> >
> > | Length  | CD↓     | HD↓     | IoGT↑   | G-Score↑ | Novel↑  |
> > |---------|---------|---------|---------|----------|---------|
> > | [0, 30]   | 0.1898  | 0.5131  | 0.7356  | 3.9234   | 56.25%  |
> > | [31, 70]  | 0.2001  | 0.5704  | 0.7021  | 3.4935   | 61.76%  |
> > |  [70, ~) | 0.2093  | 0.5924  | 0.6759  | 3.1133   | 69.34%  |
> >
> > The results indicate a clear trend: as model complexity increases, there is a decline in geometric accuracy metrics (e.g., Chamfer Distance increases). Interestingly, however, the Novelty metric shows an upward trend. We believe the probable explanation is that as model complexity rises, defining the object requires a significantly larger number of SSR commands.
> >
> > __Geometric Fidelity:__ The increased sequence length makes the generation task harder, increasing the difficulty of perfectly matching the target Ground Truth (leading to lower geometric scores).
> >
> > __Novelty:__ Conversely, the increased number of SSR commands introduces more degrees of freedom in the generation process. This enhances the diversity of the generated models, thereby boosting the Novelty score.

---

> > > ### Comment · Reviewer_AAaN · 2025-11-24
> > >
> > > Thank you for your response. I have one follow-up question: Have you evaluated your model on the CADPrompt [1] benchmark? I’m interested in seeing how your model performs across different benchmarks.
> > >
> > >
> > > [1] https://arxiv.org/pdf/2410.05340

---

> > > > ### Author Response · Authors · 2025-11-25
> > > > **Response to Reviewer AAaN on CADPrompt benchmark**
> > > >
> > > > Dear Reviewer AAaN,
> > > >
> > > > We sincerely thank you for your positive response. We add an evaluation of Seek-CAD on the CADPrompt benchmark (200 test samples). When utilizing the CADPrompt benchmark for this experiment, we exclude the specific field descriptions related to CADQuery from the original prompts. The results are presented in the table below:
> > > >
> > > > | CD↓     | HD↓     | IoGT↑   | G-Score↑ | Novel↑  |
> > > > |---------|---------|---------|----------|---------|
> > > > | 0.1986  | 0.5377  | 0.8012  | 4.1412   | 56.72%  |
> > > >
> > > > Thanks again for your feedback! We hope these clarifications could help address your concerns and allow for re-evaluating of our work. If you have any further questions, we would be delighted to continue the discussion.

---

> > > > > ### Comment · Reviewer_AAaN · 2025-11-25
> > > > >
> > > > > Thank you for addressing my concern. I have updated my score.

---

### Official Review · Reviewer_vupC · 2025-10-31

**Soundness:** 3
**Presentation:** 3
**Contribution:** 3
**Rating:** 6
**Confidence:** 4

**Summary:**

This paper presents Seek-CAD, a training-free framework for generating 3D parametric CAD models from text. It uses a locally deployed LLM (DeepSeek-R1) to generate code following a novel SSR design paradigm. A key innovation is a self-refinement loop where step-wise renders of the CAD model are evaluated by a VLM (Gemini-2.0) against the LLM's Chain-of-Thought reasoning; the resulting feedback iteratively improves the code. The authors also contribute a new 40k-sample dataset based on the more complex SSR paradigm. Experiments show Seek-CAD outperforms existing methods in geometric fidelity and text alignment.

**Strengths:**

1.The use of step-wise visual renders paired with the LLM's Chain-of-Thought for feedback is new. This provides a richer, more granular signal for refinement than methods using only the final render.

2.The proposed SSR triple and the CapType reference mechanism enables the generation of complex CAD models beyond the limitations of prior "Sketch-Extrude" methods.

**Weaknesses:**

1.The paper mentions that models failing to compile are excluded from metric calculation. A more detailed analysis of the reasons for compilation failures would be insightful.

2.While the CapType mechanism is innovative, the description in the appendix mentions that when refinement commands involve primitives not identifiable by CapType, those primitives are simply excluded. How often does this happen in the dataset/generation? Does it lead to models that are missing intended refinements?

3.The RAG corpus has 10,000 samples. Was an ablation study performed on the size of this corpus? Is there a point of diminishing returns, or could a smaller corpus suffice?

**Questions:**

Beyond compilation failures, what are the most common types of geometric or logical errors that the refinement loop fails to correct?

---

> ### Author Response · Authors · 2025-11-18
> **Response to Reviewer vupC**
>
> We are truly grateful for the reviewer’s instructive comments. Your questions are insightful and target key parts for deeper analysis. Our further clarifications are as following:
>
> __1. On Weakness 2 about CapType mechanism__
>
> We thank the reviewer for acknowledging the innovation of our CapType mechanism and for raising this very insightful question about its limitations. While we did not track the exact frequency of this exclusion during dataset parsing/generation, we can confirm that the CapType mechanism may occasionally miss certain edges, such as those created by intersections, which can lead to some intended refinements not being captured. In typical 3D design software like SolidWorks, users perform refinement operations by directly selecting the displayed B-Rep entities (faces, edges, and so on). However, B-Rep information is not explicitly represented in the construction history, meaning that prior datasets were fundamentally unable to support such refinement operations.
>
> Hence, Our goal with CapType is not to handle every possible case, but to support as many practically relevant situations as possible by bridging the gap between construction history and geometric references. Even with these limitations, our dataset already supports modeling examples of practical engineering complexity, as shown in Figure 7, marking a substantial improvement over previous data paradigms.
>
> __2. On Weakness 3 about the size of RAG corpus__
>
> Thank you for this excellent suggestion for a new experiment. To be honest, we did not include a full ablation on the corpus size in the manuscript.
>
> However, in our preliminary experiments, we did test this. We initially tried smaller corpus of 400 and 3,000 samples and found that their performance was surprisingly close to the 10,000-sample corpus. This led us to a key insight: the richness (or diversity) of the commands within the corpus is far more important than the total number of samples.
>
> The role of the corpus in our framework is primarily to serve as a reference for DeepSeek-R1, demonstrating how to generate code that strictly adheres to our defined SSR paradigm. For example, if a “fillet” command is required, but the RAG corpus contains no examples of a “fillet” being used, DeepSeek-R1 is put in a difficult position:
>
> (1) It may avoid using “fillet” entirely.
>
> (2) Or, it may try to generate a “fillet” command based on its own general knowledge base. This generated code will almost certainly not follow our specific SSR-based definition, leading to a compilation failure.
>
> We understand this is an important detail, and we will add a detailed discussion of this finding to the revised version of our manuscript.
>
> __3. On Weakness 1 and Question about “what are the most common types of geometric or logical errors that the refinement loop fails to correct? (Beyond compilation failures)”__
>
> Thank you for this question. The most common uncorrected errors can be classified into two main categories:
>
> __CapType Topological Failures:__ This is caused by the current limitations of our CapType mechanism, as discussed in our response to Weakness 2. If a logical error is related to a primitive that CapType cannot reference (e.g., "the fillet on that intersection is missing"), the loop cannot currently fix it because the system has no mechanism to point to that specific location.
>
> __Positional Constraint Errors:__ This category includes geometric errors related to constraints like "tangency" or "orthogonality" between primitives. As we state in our limitations, the cause for this is the weaker computational ability of DeepSeek-R1-32B-Q4. Even if the VLM provides correct feedback, the 32B model sometimes struggles to generate the precise parameters to satisfy these constraints. However, we have observed in further experiments that this situation is significantly improved when using the larger DeepSeek-R1-671B model. We will provide more detailed analysis in the revised version.

---

### Official Review · Reviewer_TUi8 · 2025-11-03

**Soundness:** 3
**Presentation:** 2
**Contribution:** 3
**Rating:** 6
**Confidence:** 5

**Summary:**

Authors proposed seek-cad, a training free approach for generating CAD code from textural input. It formulate data in a novel SSR template, and use a captype reference to effectively support chamfer and fillet operations on new surfaces introduced by CAD operations. The RAG system is also novel with an additional one round of step-wise visual feedback in the code refinement stage showing to help improve generation quality.

**Strengths:**

The proposed training-free approach is novel and the first few works that explored this direction. The SSR data format with captype reference is also more general and applicable to real-world scenarios than simple sketch-and-extrude. SVF is also a nice solution for incorporating step-vise visual feedback into the system and enable the model to verify each step in the built process. Evaluation on the new dataset demonstrate the improvement of seek-cad.

**Weaknesses:**

Writing and paper layout can be improved. Figure 1 is too small, and SSR definition is in the later paragraph whereas a lot of reference to it is at the front. Overall, this makes reading the paper difficult than it should be.

Evaluation is done entirely on the authors’ new SSR dataset. There is no comparison to previous methods on existing public CAD data like DeepCAD / Omni-CAD / WHUCAD. Figure 7 and 8 shows their dataset is much more complex than DeepCAD, this raise the concern that metric improvement could come from the 10,000 more complex RAG data, e.g better novelty than other methods.

Authors did not clearly explain how the test set is different from training set. E.g what kind of deduplication or similarity filter was applied.

Authors do not provide concrete implementation details for Eq. (1) and Eq. (2) — only high-level descriptions of what those equations represent conceptually. This makes reproducing the work fairly complicated.

**Questions:**

(1) Why not use a single vllm model like Qwen-VL or InternVL. The proposed design using DeepSeek for text and Gemini 2.0 for visual is fairly complicated. Is there a particular reason why authors use this pipeline?

(2) How is the dataset constructed? What method is used to avoid data leaking from training to test set? Does it have the clearance from OnShape to be allowed to be publicly released?

(3) How important is the RAG data? Does the increase in complexity help?

(4) Is it possible to use exsiting public CAD data as the RAG data and see how it compares to baselines? This seems like the fair way to compare without the results been affected by the new dataset.

(5) Please provide implementation details in the paper for reproducible results.

---

> ### Author Response · Authors · 2025-11-21
> **Response to Reviewer TUi8 Part(1/2)**
>
> We are deeply grateful for the reviewer’s constructive comments. We appreciate the time and effort you dedicated to reviewing our paper and recognize the novelty of our training-free approach, the generalizability of the SSR format with CapType, and the effectiveness of our Step-wise Visual Feedback with CoT. Your feedback regarding the presentation and experimental design has been instrumental. Our responses to your specific questions are provided below.
>
> __1. On Weakness 1 about paper layout__
>
> We fully accept this valid criticism. In the revised manuscript, we will make the following improvements:
>
> __Enlarged Visualization:__ To address the legibility of Figure 1, we will include enlarged, high-resolution visualizations of its key design modules in the Appendix.
>
> __Reorganize the sections:__ We will move the definition of the SSR Paradigm (currently Sec 4) to appear before or immediately within the framework description (Sec 3), ensuring the reader understands the core data structure before diving into the generation pipeline.
>
> __2. On Question 1 about Why not use a single vllm model like Qwen-VL or InternVL. The proposed design using DeepSeek for text and Gemini 2.0 for visual is fairly complicated. Is there a particular reason why authors use this pipeline?__
>
> We thank the reviewer for raising this insightful question regarding our architectural choices.
>
> Our pipeline design was primarily inspired by the exceptional Chain-of-Thought (CoT) reasoning capabilities of DeepSeek-R1. At the inception of this project, DeepSeek-R1 demonstrated superior CoT performance among available LLMs. We found its ability to explicitly articulate the step-by-step logic of CAD modeling, such as "first draw a line, then draw an arc…", which aligns perfectly with the sequential nature of feature-based CAD design.
>
> This specific CoT capability serves as the foundation for our key architectural contribution: the Step-wise Visual Feedback (SVF) mechanism guided by CoT. The necessity of this design is empirically validated by our Ablation Study (Table 3). Specifically, Model G (which removes the CoT guidance from the SVF loop) shows a significant performance drop (e.g., G-Score drops from 3.8409 to 3.6120). This confirms that DeepSeek-R1's strong CoT reasoning is indispensable for the effectiveness of our current pipeline.
>
> We acknowledge the rapid evolution of Large Language Model technology. It is entirely possible that the very latest unified VLLMs could now achieve comparable results. Exploring such unified architectures is a promising direction that we intend to investigate in future work.
>
> __3. On Weakness 3 & Question 2 about How is the dataset constructed? What method is used to avoid data leaking from training to test set? Does it have the clearance from OnShape to be allowed to be publicly released?__
>
> Thank you for these questions. Our dataset is constructed based on the original FeatureScript definitions of CAD models provided in the ABC dataset, which represent the procedural generation process of each model. We further use the Onshape API to assist in analyzing these FeatureScript definitions. We carefully reviewed Onshape’s Terms of Use and confirmed that our workflow does not violate them. Essentially, we only parse ABC’s publicly available FeatureScript definitions with the help of the API; we do not collect any user-generated data from Onshape. Therefore, publicly releasing our dataset does not pose any licensing or compliance issues.
>
> Regarding data leakage, we observed that applying the deduplication procedure provided by DeepCAD still leaves a considerable amount of duplicated content. This issue also exists in the DeepCAD dataset itself: even after deduplication, a non-trivial portion of the training and test sets remain duplicates. One reason is that the same CAD shape can correspond to multiple distinct construction sequences, causing sequence-level deduplication to be insufficient.
>
> To address this, we further perform deduplication based on pairwise image similarity of the resulting CAD shapes. For samples whose similarity exceeds a threshold (we use 0.95), we retain only one instance in the final dataset. We find that this method significantly reduces data leakage and substantially improves dataset diversity.
>
> __4. On Weakness 4 & Question 5 about Please provide implementation details in the paper for reproducible results?__
>
> We sincerely thank the reviewer for pointing out this important issue. We acknowledge that the implementation details in the current manuscript are indeed not concrete enough.
>
> In the revised version, we will update the manuscript to include more detailed implementations to clarify our whole pipeline. Furthermore, we will make the code public upon acceptance to ensure complete reproducibility.

---

> > ### Author Response · Authors · 2025-11-21
> > **Response to Reviewer TUi8 Part(2/2)**
> >
> > __5. On Weakness 2 about dataset used & Question 4 about more comparisons on the existing public dataset.__
> >
> > We fully understand the reviewer's concern about fairness, but we respectfully argue that testing on DeepCAD/Omni-CAD would not accurately measure our contribution. DeepCAD and Omni-CAD are restricted to the Sketch-Extrude (SE) paradigm. They explicitly exclude the complex features (chamfer, fillet, shell) that constitute the core innovation of our SSR paradigm. It would fail to demonstrate the model's ability to handle the topological complexities (CapType) that prior methods simply cannot model.
> >
> > Besides, we closely also follow the development of WHUCAD and acknowledge it as a valuable contribution containing complex CAD models. However, the currently public WHUCAD dataset is provided in B-Rep (Boundary Representation) format. Converting this B-Rep data into our sequential SSR Paradigm (which requires construction history and specific feature operations) is a non-trivial Inverse-CAD task requiring significant adjustments. Therefore, we could not directly use it for this study. We will include a discussion of WHUCAD in the Related Work section of our revised manuscript and plan to explore converting it to the SSR paradigm in future work.
> >
> > To ensure fairness, we did not just run Seek-CAD with RAG corpus (10, 000 CAD models) based on our complex SSR dataset. We also adapted the baselines with the same RAG corpus (10,000 CAD models). As shown in Table 1, all methods were evaluated on the same test set (500 complex CAD models from our SSR dataset).
> >
> > Furthermore, we value your suggestion to include comparisons on public datasets. We are happy to provide a new set of comparative experiments conducted on the DeepCAD dataset. For this experiment, we replaced our local RAG corpus with 4,000 CAD models collected from DeepCAD and another 300 CAD models (different from 4,000 CAD models in the local RAG corpus) as the test set. The comparative results are as follows:
> >
> > | Method        | CD↓     | HD↓     | IoGT↑   | G-Score↑ | Novel↑  |
> > |---------------|---------|---------|---------|----------|---------|
> > | 3D-PreMise    | 0.2079  | 0.5823  | 0.7375  | 3.3192  | 48.89%  |
> > | CADCodeVerify | 0.2035  | 0.5745  | 0.7531  | 3.6452   | 51.18%  |
> > | **Seek-CAD**  | **0.1811** | **0.5231** | **0.8095** | **4.0604** | **54.78%** |
> >
> > These experimental results demonstrate that even on the simpler SE-paradigm dataset, Seek-CAD continues to outperform the baselines across evaluation metrics. This validates the feasibility and robustness of our framework, confirming that its effectiveness is not solely dependent on our proposed SSR dataset.
> >
> > __6. On Question 3 about How important is the RAG data? Does the increase in complexity help?__
> >
> > The RAG data is absolutely critical. Please refer to our Ablation Study (Table 3, Model A). When we removed the local CAD corpus (w/o RAG), model A completely fails to generate valid CAD codes (Pass@k rate dropped to 0.0%). Since Seek-CAD is training-free, the LLM has never "seen" our specific SSR data format. The RAG component provides the necessary few-shot examples to teach the LLM the syntax and structure of the SSR paradigm in-context. Without it, the model cannot generate valid CAD code.
> >
> > Yes, the RAG data does increase in complexity help. If we only used simple RAG data (like the SE-based DeepCAD), the LLM would be restricted to mimicking simple SE paradigm, which cannot support fillet, shell, chamfer operations, reducing the complexity of the generated CAD models.

---

### Author Response · Authors · 2025-12-02
**Authors' Final Comments on Rebuttal Summary**

We sincerely thank the reviewers for their time and constructive comments, and we appreciate their recognition of our paper’s novelty.
To address the reviewers' questions and concerns, we provided comprehensive responses and conducted extensive additional analyses during the rebuttal period, including:

__(1) Added performance comparisons on the public DeepCAD dataset__

__(2) Evaluated Seek-CAD on the CADPrompt benchmark__

__(3) Further explained the CapType mechanism and the critical role of the RAG corpus__

__(4) Provided more details of the dataset construction__

__(5) Added the analysis of the costs between fine-tuning strategies (CAD-Llama) and our training-free strategy (Seek-CAD)__

__(6) Analyzed the most common failure cases (CapType errors and Constraint errors)__

__(7) Provided more explanations regrading the reliability and quality of VLM feedback__

__(8) Explained the reason for choosing DeepSeek-R1+VLM as the pipeline__

Initially,  we got scores of (6,6,4,4). We are pleased to report that following an intensive and fruitful discussion with Reviewer AAaN, the Reviewer AAaN's score was __upgraded__ from __4 to 6__, which took place prior to the OpenReview leak. Due to the premature termination of the discussion period caused by the OpenReview leak, we regrettably did not have the opportunity to receive feedback or engage in further discussion with the remaining three reviewers after posting our rebuttal. Overall, our scores were updated from (6, 6, 4, 4) to __(6,6,6,4)__.

---

### Meta-Review · Area_Chair_HDzo · 2026-01-01

**Summary:**

Reviewer TUi8

*) Criticism about writing and formatting
The authors addressed this in the rebuttal. The proposed changes seem reasonable.

*) The evaluation is limited and only performed on a self collected dataset
The authors address this in the rebuttal by testing on a curated subset of existing datasets. The provided test is again not a standard test and the results are inconclusive.

*) Authors did not clearly explain how the test set is different from training set.
This is addressed in the rebuttal, but it is unclear how much the sets overlap. It is possible that the method can greatly benefit from near duplicates. The rebuttal shows the authors took some care to avoid this, but this is not convincing.

*) Missing implementation details
addressed in the rebuttal

The reviewer may keep, reduce, or increase the score. I estimate the reviewer would keep the score of 6, but important aspects of the paper are unclear.

Reviewer vupC:
*) The paper mentions that models failing to compile are excluded from metric calculation. A more detailed analysis of the reasons for compilation failures would be insightful.
This is addressed in the rebuttal

*) While the CapType mechanism is innovative, the description in the appendix mentions that when refinement commands involve primitives not identifiable by CapType, those primitives are simply excluded. How often does this happen in the dataset/generation? Does it lead to models that are missing intended refinements?
This is addressed in the rebuttal

*) Missing ablation study on the number of RAG samples
The authors do not provide this in the rebuttal

I would predict the reviewer would keep the score of 6, because the RAG ablation study was not provided.

Reviewer AAaN:

There was a discussion between the authors and the reviewer about the VLM feedback quality and the compilation failure rate. The reviewer decided to update the score to 6.

Reviewer fiYb:

*) It is unlcear how often the VLM helps and how often it hurts.

This was partially addressed in the rebuttal.

*) The reviewer was concerned about the multi-stage refinement

The rebuttal confirms that this is a problem, but claim the system still works overall.

I predict the reviewer would either keep or increase the score so the expected value is 5

**Reviewer Concerns:**

See above.

**Reviewer Scores:**

I predict the reviewer scores of 6, 6, 6, 5, and recommend acceptance as a poster.

The decision may be revised downwards since there are some critical issues about the performance unclear as discussed above. Still, I believe it would make a solid poster.

---

### Decision · Program_Chairs · 2026-01-26

Accept (Poster)